# Mobile Clustering Scheme for Pedestrian Contact Tracing: The COVID-19 Case Study

**DOI:** 10.3390/e23030326

**Published:** 2021-03-10

**Authors:** Mario E. Rivero-Angeles, Víctor Barrera-Figueroa, José E. Malfavón-Talavera, Yunia V. García-Tejeda, Izlian Y. Orea-Flores, Omar Jiménez-Ramírez, José A. Bermúdez-Sosa

**Affiliations:** 1Communication Networks Laboratory, CIC-Instituto Politécnico Nacional, Mexico City 07738, Mexico; 2SEPI-UPIITA-Instituto Politécnico Nacional, Mexico City 07740, Mexico; 3Telematics Section, UPIITA-Instituto Politécnico Nacional, Mexico City 07738, Mexico; jmalfavont1300@alumno.ipn.mx (J.E.M.-T.); iorea@ipn.mx (I.Y.O.-F.); mjimenezr1200@alumno.ipn.mx (O.J.-R.); jbermudezs@ipn.mx (J.A.B.-S.); 4Basic Sciences Section, UPIITA-Instituto Politécnico Nacional, Mexico City 07340, Mexico; ygarciat@ipn.mx

**Keywords:** building access, mobile clustering scheme, tracking of pedestrians, RFID systems, control of virus propagation

## Abstract

In the context of smart cities, there is a general benefit from monitoring close encounters among pedestrians. For instance, for the access control to office buildings, subway, commercial malls, etc., where a high amount of users may be present simultaneously, and keeping a strict record on each individual may be challenging. GPS tracking may not be available in many indoor cases; video surveillance may require expensive deployment (mainly due to the high-quality cameras and face recognition algorithms) and can be restrictive in case of low budget applications; RFID systems can be cumbersome and limited in the detection range. This information can later be used in many different scenarios. For instance, in case of earthquakes, fires, and accidents in general, the administration of the buildings can have a clear record of the people inside for victim searching activities. However, in the pandemic derived from the COVID-19 outbreak, a tracking that allows detecting of pedestrians in close range (a few meters) can be particularly useful to control the virus propagation. Hence, we propose a mobile clustering scheme where only a selected number of pedestrians (Cluster Heads) collect the information of the people around them (Cluster Members) in their trajectory inside the area of interest. Hence, a small number of transmissions are made to a control post, effectively limiting the collision probability and increasing the successful registration of people in close contact. Our proposal shows an increased success packet transmission probability and a reduced collision and idle slot probability, effectively improving the performance of the system compared to the case of direct transmissions from each node.

## 1. Introduction

Smart City is a general term used to represent an information system installed in certain intelligent environments to disseminate information regarding the operation of the city and the quality of life of the residents [1], such as information about the water supply, traffic conditions, dangerous leaks, cultural and sports events, health-related information, updates about pollution, and climate-related data among many others. In pandemic times, smart city environments would allow having detailed monitoring of contagion hotspots by following close encounters of potential sick individuals with other people in specific periods and locations. With this information, governmental authorities can isolate specific individuals who might be at risk of infection without affecting the rest of the population, effectively avoiding the complete lockdown of the city and reducing the economic impact of the epidemic.

The outbreak of a new virus that emerged in the city of Wuhan, central China in December 2019, has crossed all borders. The virus was named Severe Acute Respiratory Syndrome Coronavirus 2 (SARS-CoV-2 for short) by the International Committee on Taxonomy of Viruses. Consequently, COVID-19 (Coronavirus Disease 2019) is the illness that it causes [2]. On 8 February 2021, the Worldometer COVID-19 Data reported 106,952,085 cases with COVID-19 and confirmed 2,334,563 deaths worldwide [3].

It is generally known that SARS-CoV-2 is primarily transmitted via airborne, through aerosols emitted during speech, sneeze or cough, by infected individuals [4]. Aerosols containing SARS-CoV-2 remain suspended in the air for hours [5], and may potentially be inhaled by others. Therefore, for reducing the risk of transmission, physical distancing of 2 m is considered as an effective protection only if everyone wears face masks in daily life activities [6]. However, small particles with viral content may travel in indoor environments, covering distances up to 10 m starting from the emission sources [6]. Moreover, the temperature, lighting, and humidity of the environment are abiotic factors that influence the inactivation of COVID-19.

It is reported that exposure of the SARS-CoV-2 virus for 30 min at 34${}^{\circ }$ C, in the absence of humidity, is sufficient to damage the structure of the spike protein [7], which binds to ACE2 (Angiotensin-converting enzyme 2) the receptor for SARS-CoV-2 [8]. In contrast, at 22${}^{\circ }$ C, the spike protein structure remains unaffected. Regarding exposure of the virus to sunlight, about 90% or more of SARS-CoV-2 virus is inactivated after being exposed for 11--34 min of midday sunlight in Mexico and in many other world cities during summer. On the contrary, the virus is active for a day or more in winter [9].

UV light is divided into three classifications: UVA (320–400 nm), UVB (280–320 nm), and UVC (200–280 nm). UVC is absorbed by RNA and DNA bases [10], which is effective for inactivating SARS-CoV-2 [11]. The monitoring of the above-mentioned abiotic factors can help us to determine the pedestrian occupancy in a building or space, and implement precautionary measures to reduce risks of getting COVID-19.

In current times, the COVID-19 epidemic has shown the devastating effects of poor infection monitoring and tight and generalized quarantine conditions in different cities regarding the economy and general health of their population, due to prolonged isolation, lack of social interaction, and sunlight, among others. Furthermore, experts warn that this current epidemic is only one of many others to come in the near future. Hence, we suggest being ready for future emergencies by having the computational tools to avoid catastrophic scenarios as the one lived worldwide in 2020–2021. To this end, contact tracing is a critical tool to stop the spreading of such diseases [12,13,14].

Our monitoring system employs devices with Radio Frequency (RF) communication capabilities to periodically send a beacon signal to form mobile clusters (pedestrians are usually moving in specific trajectories and the clusters move accordingly) with individuals inside their radio of interest, i.e., within the distance of possible contagion. This radio can be modified according to the abiotic factors monitored at certain times during the day. Even if worldwide safety guidelines clearly state to keep social distancing of at least two meters, in many cases, these recommendations have not been respected as has been documented throughout the world [15,16]. This is the main reason for the increment in confirmed cases at different periods of this pandemic. In some cases, the 2-m social distance has not been respected due to cultural, religious, social, and/or commercial reasons where people continue to gather in special dates that they are used to celebrate, despite police installing sanitary filters and trying to separate people by force, as reported in [15]. Furthermore, as mentioned in [17], urban agglomerations, where the social distancing is not fully respected, are the centers at highest risk during a period of a pandemic. Indeed, more than 90% of COVID-19 clusters are associated with densely populated urban agglomerations and megacities in the world [17]. This is can be explained in part by the fact that people have not respected the 2-m distancing on their daily activities since cities are the economic and financial motor of many developing countries and many activities cannot be completely stopped, workers have to travel, in many cases using the public transport system where close contacts cannot be avoided as noted in [18]. Building on this, we believe that using a short-range (lower than 2 m) clustering scheme for communication among pedestrian’s devices for contact tracing applications entail important benefits.

Although ad-hoc wireless sensor networks could be designed based on Bluetooth technology (see, e.g., [19]), we find some issues that limit its application to the tracking of pedestrians in a pandemic scope. In the first place, its high power consumption: establishing a wireless Bluetooth link at a distance of 10 m (Power Class 2) or 1 m (Power Class 3) requires 4 dBm or 0 dBm, respectively. In this work, we designed a wireless device that employs only –17.5 dBm for establishing a reliable wireless link, which is far lower than the previous transmitting powers. In the second place, the pairing of devices: for the correct pairing, the Bluetooth devices should be close enough depending on their transmission power and the level of security. It is observed that the higher the level of security, the larger the average time needed to establish a link (see, e.g., [20]). This may require a few seconds, which is far larger than the microseconds used to transmit data between our RF devices. In the third place, security: though Bluetooth employs an authentication protocol to connect two devices, as well as encryption for transmitting data, it is not difficult to attack a Bluetooth device (see, e.g., [21]), which compromises the protection of information of the pedestrians.

Then, our proposal focuses on nodes inside the mobile cluster to share their ID in order to detect the people that was in close interaction with each individual throughout the day. Hence, whenever an individual is confirmed to be infected (by a trusting health authority), the city services can easily detect the people that could be in potential infection danger by determining the identity of the people that were close to this infected person in the previous days. Thus, these people in potential danger can be isolated as well as the people that were in touch with them and so on and so forth.

The use of mobile clusters aims at making efficient utilization of resources, increasing the success transmission probability of nodes. For mobile applications, Cluster Heads (CHs) can transmit the close contact information from the nodes inside the cluster directly to a base station or access point whenever a close contact of less than the *contagion range* occurred, avoiding the need of storing this data in the device and periodically (once a day, in some cases) accessing remote databases. For the case of using specific devices, with much lower computational and storage resources than mobile phones, that cannot access databases or maintain many contact IDs for many days, the mobile clustering scheme also reduces replicated data—due to selecting a single Cluster Head in charge of receiving and concentrating the information for the rest of the nodes in the cluster and concentrating this data into a single transmission to the city/institution administrator, making efficient use of the scarce RF and computational resources as well as reducing energy consumption compared to the case where each node constantly transmits data to the disease control agency. In a smart city environment, many *contagion control points* can be distributed in strategic parts of the city, such as schools, hospitals, access to massive transport services, and others to have close control of potential pedestrian contagion and swiftly take action to reduce and limit infection spreading.

By providing an individual monitoring system, like the one proposed in this work, using either a smartphone or specific devices provided by different institutions like universities, hospitals, etc., it is possible to maintain open many buildings and commerces, thus reducing the risk of contagion. For instance, universities can provide RF devices to all their students, professors, administrative personnel, etc., in order to remain open and isolate only both sick individuals and all the people that were in touch with them in specific periods and locations. The benefits of such specific devices are the additional security and privacy compared to using mobile phones. Indeed, in case of a cyber attack, hackers could obtain much more information from smartphones while monitoring devices can only provide IDs, which are only known by the appropriate authorities. It is important to note that the use of applications in the mobile phone is much easier than a device dedicated to contact tracing. However, as mentioned in [22], there are many privacy concerns related to these applications, especially about systems based on tracking the geographical location of app users. This is one of the main deterrents for installing such tracking systems in personal mobile phones, especially in countries where governmental entities are not trusted (whether this appreciation is justified or not) by the general public or the official measures to combat the COVID-19 pandemic does not consider accurate contact tracking as the main tool to reduce strict confinement conditions. Such is the case of Latin America and Africa (note that there are no official applications listed in [22] for countries like Mexico, Guatemala, Honduras, Angola, Egypt, and many more), where also the COVID-19 epidemic has high contagion levels [23]. As such, applications in mobile phones are not a practical or preferred solution for all countries and cities to keep contact tracing information. For these cases, the use of a personal device provided by the university or factory or hospital, or any other place where a high concentration of people is expected, with no other information than a specific ID could be a much-accepted solution. It would also reduce cyber-attacks aimed at obtaining personal information from mobile phones. Hence, we argue that the use of specific devices may not be adopted as a general solution for contact tracing at the country level, but it may provide an effective and accepted solution for specific enterprises, commerce locations, and governmental and health entities that would allow an anticipated reopening solution to mitigate the economic and social negative impact of generalized lockdowns. For countries where mobile phone applications are well accepted, the mobile clustering scheme and results can be relevant and provide further insights on the performance of such solutions.

Some monitoring systems aim at keeping a close register regarding the specific individuals that enter a certain location. For instance, in Mexico City and China [24], a governmental application using QR codes at the entrance of buildings and commerces, requires clients to scan the QR code in order to register the time at which a person was inside the facilities. Hence, if a certain person is diagnosed with COVID-19, such a system determines the people that were at the same time and place as the infected individual, which are at a high risk of contagion. However, this system is not accurate enough. Indeed, the fact that other people were inside the same building at the same time than an infected individual does not imply that they were at any time at contagion distance from each other: they could have been at different locations inside the building, never crossing paths, and the system could warn a higher number of people than needed to keep them in quarantine. In addition, the efficacy of such a system is compromised considering that the authorities implemented this service at the end of 2020 while confirmed COVID-19 cases increased exponentially in early 2021.

Other applications that use mobile phones to detect people in close range produce redundant information by requiring all of these nodes to transmit the list of people (mobile phones) that were close to them. This redundant information entails high energy consumption and implementation costs by using resources from the cellular system that are expensive. Conversely, our proposal keeps an accurate register of people that were in actual contagion distance from each other inside and outside buildings. Furthermore, a single user in a group of people is selected to convey this information to the disease control agency reducing the number of transmissions and redundancy.

The main contributions of this paper are:We propose a novel mobile clustering scheme to efficiently gather information of close contact with other people in pedestrian scenarios.We design and propose a specific communication protocol to implement a mobile clustering scheme.We mathematically model the proposed mobile clustering scheme to study the system performance in different scenarios by providing a teletraffic analysis of the dynamics of contact instants and encounters in pedestrian scenarios.We design and develop our own RF device with the specific objective of maintaining a record of people in close contact under the mobile clustering scheme.

Moreover, the derived mathematical model can be easily extended and adapted to different scenarios and cities in order to provide other services related to smart city applications in non-pandemic times, like intelligent access control to buildings or public transportation.

The rest of the paper is organized as follows: first, we present relevant works in the context of contact tracing systems. Then, the system model is described in detail. Following this, Section 4 presents the main operation of the mobile clustering protocol. Then, the pedestrian mobility pattern is characterized in Section 5. The mathematical model is derived in Section 6 where all the variables and assumptions are explained. In Section 7, we present the design and performance of the node developed for contact tracing purposes. The paper ends by presenting the most relevant results and conclusions.

## 2. Related Work

Contact tracing is an essential tool to improve the health of people and to reduce the economic impact of a pandemic such as the one derived by the SARS-CoV-2 virus [13]. To this end, there have been many efforts to develop digital tools to keep track of contact among people to rapidly detect possible contagion hot spots.

For example, the study in [12] describes the strict contact tracing scheme used in South Korea that uses data from the Global Positioning System (GPS), credit card transactions, and video surveillance among other systems in order to reduce the contagion cases of COVID-19, clearly showing the benefits of such policies.

Regarding the digital contact tracing tools, there are many different applications with government support in certain regions as mentioned in [22], are centralized protocols concentrating all personal data (geo-localization) in state institutions. For instance, Israel approved the secret service to use surveillance measures to access information of users connected at different networks, which can have many potential privacy issues. Decentralized protocols like the one developed by Covid Watch, the CEN Protocol, based on Bluetooth Low Energy (BLE) using proximity among cellular phones to detect potential contagion cases. In this regard, the Pan-European Privacy-Preserving Proximity Tracing (PEPP-PT) project (a combination of centralized and decentralized approaches) developed a BLE app aimed at detecting such close interactions and avoid state surveillance activities. Later on, different institutions criticized the PEPP-PT for lack of transparency and privacy issues [22]. Nonetheless, these decentralized approaches aim at protecting private information using anonymous keys that have no relation to the user’s identities. However, these applications do not function properly if only a small population uses the app [25], which occurs even if workers are legally required to use it [26]. However, in a closed environment such as universities or hospitals where employee access to the buildings can be conditioned on using a specific RF device just like ID is commonly required (or even IDs can be placed on such devices), our proposed device could be a better option since it does not require administrative access to the mobile phone in order to implement contact tracing and the exposure of smartphones is avoided, which can be potentially dangerous to people since mobile phones are reservoirs for various pathogens [27]. In addition, apps that use Bluetooth and GPS to estimate the distance may over-report interactions leading to a high number of false positives [22]. By contrast, the development of a specific device has the advantage of fine-tuning the *contagion range* according to the specific needs thorough the careful design of antennas, amplifiers, and filters. Indeed, for COVID-19, the official recommendation is to avoid close contact of less than 1.5 or 2 m, but variations of this virus or for other viruses in the future, this social distancing can be different and the RF device can be designed accordingly, while GPS and Bluetooth systems cannot easily do. In this work, we propose the use of both approaches, based on apps on mobile phones and specific RF devices, in order to offer a general solution for contact tracing efforts in the sense that the mobile clustering scheme provides an efficient data reporting in pedestrian environments.

Many papers have studied the effectiveness and uses of contact tracking applications and models [28]. For instance, in [29], the authors perform different simulations to evaluate the effectiveness of such tools. However, they do not consider the real possible interactions between pedestrians. Instead, they rely on census data which have no information on contact times in a given trajectory, like our work.

In [30], the authors consider a network-based model to calculate the infection spread in a close population. Specifically, this work proposes a stochastic simulation and moment closure approximation where nodes are placed at certain distances to calculate the probability that two connected nodes get infected. A similar approach is presented in [31], where the authors study the dynamics of contagions using a stochastic epidemic model based on the embedded non-stationary Galton–Watson process. However, they do not consider the times that nodes are in contact with each other in a pedestrian movement pattern nor the dynamics of the nodes in such an environment.

In [32], the authors discuss privacy and trade-offs of such contact tracing applications. To this end, we propose the use of specific nodes that do not contain personal information, except for an ID (assigned by the government, institute, enterprise, or health authority) and geo-localization data that present minimum risk in case of a cyber attack.

As mentioned in [33], many applications use Bluetooth signal strength to calculate the distance among people and duration of such contact, while others rely on geo-localization data using GPS information to determine the proximity of individuals [34,35,36,37]. However, we propose to use direct transmissions to nodes in the proximity to clearly identify the people in close contact to others, which we believe would render more accurate proximity results since it implicitly considers the contact time that nodes were in contact with each other instead of instant contacts provided by Bluetooth and GPS signals. Additionally, by using the node developed and designed in our work, the contagion radius can be easily selected according to the specific disease phenomena, making our system general for current and future pandemics. In addition, the proposed mobile clustering scheme can inform of potential infection cases in hours, when the CHs report their data to the sink point when they are placed in strategic locations, or minutes if the cellular system is used. As mentioned in [38], for a contact tracing tool to be effective, it has to report potential infection cases in less than one day, contrary to GPS-based solutions that have to be retrieved from GPS records that can take many days to analyze.

## 3. System Model

In this section, the basic assumptions and system variables of the mobile clustering scheme are presented in detail. Our work is focused on pedestrian movement in outdoor conditions, where people are walking towards a building, institution, or commercial location. Specifically, we consider the case of students entering the facilities of the National Polytechnic Institute in Mexico City. In Figure 1, we present the general operation of the proposed system. The system operates by clustering neighbor nodes (associated with pedestrians) moving with a certain trajectory. Given the dynamics of the users, nodes select their role as either Cluster Head or Cluster Member according to the protocol described in Section 4. The nodes begin the packet transmission after an initialization packet is received that is periodically transmitted by a beacon localized at strategic points. In our case, the beacon emitter is placed at the street leading to the entrance of the National Polytechnique Institute (left part of Figure 2). (For other applications, beacon and sink nodes can be placed at the entrance of the supermarket or the subway or hospital, among others.) While pedestrians are walking along the street, they form clusters (depicted by red hexagons) and act according to their role (which can also change according to the possible scenarios described below), i.e., CMs transmitting their packet to their associated (nearest) CH. The approximated distance is calculated by the strength of the signal emitted by each CH. When the CHs detect the sink node, they transmit the gathered information while the CMs remain silent after successful transmission to their respective CH. The sink node stores this information (time and place that the nodes that were part of a cluster and hence were in close interaction among each other and are potentially in danger of contagion in the case that one of them receives a positive COVID-19 test result in the following days) that can be accessed by the trusted authority, e.g., government or health care institution, such that, in case of a positive test of the virus, the people in potential contagion danger can be prevented and put in quarantine. In the case of mobile phone users, the sink and beacon emitter function can be performed by the attending base station.

In this case, students typically arrive in high concentrations at certain times in the morning, just before day-time courses begin and in the afternoon when evening courses start. In order to determine the characteristics of the student’s movement, we took many hours of video in the street leading to the entrance of the institute at peak traffic periods, i.e, when a high concentration of people is attempting to enter the campus. The rationale behind this is that the proposed mobile clustering scheme aims at reducing packet transmissions and making efficient resource utilization by reducing data redundancy. To this end, a single node gathers the information from neighboring nodes. As such, this scheme is effective when there are many nodes walking in close proximity. When single nodes (pedestrians) are walking, the mobile clustering scheme has no meaningful impact compared to direct node transmissions to the sink.

### Pedestrian Trajectories

As previously mentioned, we obtained real pedestrian trajectories as the ones presented in Figure 2 (pedestrians enter from the left side of the figure). Specifically, we recorded the pedestrians accessing the National Polytechnic Institute, in Mexico City, on different days and times. In total, we used 15 different trajectories for this work. Even if we obtained many more hours of recordings, we found out that many of them could not be used due to obstacles, such as cars and buses passing through, which impede the tracking of each individual.

At this location, pedestrians follow a specific behavior in the sense that they are all walking towards the entrance of the institute (upper right of Figure 2). In addition, there are some well-defined schedules where pedestrian traffic is intense, while, at other times, there is almost no one walking through. This is a relevant characteristic of people accessing an education building, such as universities or colleges, due to strict times for the beginning and end of the courses. However, this is also true for much other indoor access, such as commercial malls, governmental buildings, and also access to public transport like the subway. Hence, we believe that the analysis presented for this specific case can be easily extended to many other scenarios.

On this basis, we consider that each person in the recorded environment has a mobile device that can communicate with its neighbors, either using a smart phone or a specific RF device (like the one developed in this work) that implements the mobile clustering scheme described in detail in the next section. Then, a node in each cluster acts as Cluster Head (CH), receiving data (ID, timestamp, location, etc.) from the rest of the nodes in the cluster who acts as Cluster Members (CMs). For smartphones, this information is available for most cases, while, for specific communication devices, this information can be provided by specific beacons placed in strategic locations informing the location and time that the user crossed a certain area. In the specific case of the National Polytechnic Institute, these beacons can be placed at the exit of bus stops and subway stations closest to the different campuses or at the streets leading to the entrance of the facilities, like the left part of Figure 2. Unlike a conventional clustering scheme [39], CMs and CHs can enter and leave their current cluster, in which case the system must adapt to the ongoing changes. Then, each CH sends its data to a sink node while the rest of the CMs remain silent after their packet has been successfully received by their CH. In this work, the sink is placed at the entrance of the Institute, but it can also be placed at strategic points in the street or can even be sent to a cellular base station, depending on the particular scenario.

## 4. Mobile Clustering Scheme

In this section, we describe the operation and main considerations of the mobile clustering scheme developed in this work.

### 4.1. Initialization Phase

When nodes are turned on or smartphones are activated in the tracking application, they become CH with probability *P* and CM with probability 1−P. This can be done when people leave their house or work and interaction with other people begins. Hence, devices do not have to be turned on at all times and smartphones also do not have to be active.

Conversely, for close communities, such as universities, the sink node can perform the beacon transmission, in order to save energy consumption at nodes. Hence, when nodes are turned on, they enter reception mode waiting for the beacon signal, as shown in Figure 3. As such, nodes would not transmit information outside the campus since the sink is placed at the entrance of the facilities; this also provides a certain level of privacy.

At this point, CHs begin transmitting a beacon signal periodically each $T_{b}$ seconds, with the transmission power necessary to reach neighbor nodes. (This transmission range has to be carefully calibrated according to the specific virus characteristics, i.e., the social distance where people can be infected.) On the other hand, CMs enter into reception mode continuously listening to the channel to receive a beacon.

If CMs do not receive a beacon after $T_{w}$ seconds ($T_{w}>T_{b}$), they assume that no CH is present and they become CH with probability *p*. The impact of the selection of both parameters *p* and $T_{b}$ in the system performance is studied later in this work.

### 4.2. Data Transmission Phase

Once that the CHs have be selected, and the beacon signal is transmitted with a timestamp, geographical information, and ID of the CH, all receiving CMs transmit their packet (with their ID, geographical information, and timestamp) with probability τ as shown in Figure 4. The reason for this is to avoid packet collisions among CMs in the current cluster. CMs only perform a single successful transmission (a single CM transmission that suffers no packet collision) to their current CH. At this point, the CH effectively registers that such CM is inside the *contagion* range at that specific time and place and will report it (with the information of the rest of the CMs) to the sink node. However, these nodes may report again if a different CH is found, which means that a new set of nodes may be present and the conditions of new contagion cases may exist.

Building on this, the mobile clustering protocol may encounter different possible scenarios that are important to study, as described below.

### 4.3. Possible Operation Scenarios

As we have noticed in the pedestrian trajectories, after a certain time that the students walk towards the entrance, the clusters reach a certain *stable phase* where the same set of nodes moves almost at the same pace resulting in the same number of nodes in the proximities of each other. A single CH and multiple CM scenario is illustrated in Figure 5. In this case, each cluster enters the normal operation described above. This is the best-case scenario since there is no need for additional packet transmissions and, after a few seconds, only the CH is periodically transmitting the beacon packet, but CMs no longer respond. However, due to the mobility of nodes, CHs and CMs can enter or leave an already formed cluster, which would trigger the following actions from the mobile clustering scheme.

(A) Multiple CMs and no CH: This is the case where the CH leaves its cluster given that this node walks slower or faster than the rest of the nodes in the cluster and CMs have not transmitted their packet, as depicted in Figure 6. In this case, after $T_{w}$ seconds, the CMs do not receive the beacon signal and assume that the CH is no longer present. Hence, each member becomes a new CH with probability *p*. The first CM to become CH sends its beacon packet, informing the rest of the nodes that he has taken the role of CH. In the case that another CM also becomes CH, when listening to the beacon packet, it returns to the CM role. At this point, the cluster enters the normal protocol operation.

(B) Multiple CHs: In this case, when two pedestrians with the role of CH encounter each other in the same *contagion range* illustrated in Figure 7, each one with its beacon transmission time (even if the beacon is transmitted each $T_{b}$ seconds, the transmissions do not occur at the same time), the first CH that transmits its beacon will remain as CH, while the other CHs become CMs, returning to a normal operation state. CHs that become CMs transmit all the gathered data retrieved at this point to the new CH.

Since our proposal is based on clustering nodes according to the distance among them, and this distance is estimated by the signal strength, there could be inaccuracy and uncertainty given by signal fading, interference, and noise in the environment. Hence, to further improve the system precision, different fuzzy techniques applied to clustering can be used, such as the one presented in [40].

## 5. Mobility Statistics

In this section, we study in detail the duration of the connection times among nodes during their trajectory towards the entrance. Specifically, using the videos taken at rush hours, we determine the times that each node remains in each cluster either as a CH or CM.

Specifically, the videos show frame by frame the position of each pedestrian that walked along the street, giving us the location of each potential node (at this point, pedestrians are not equipped by RF nodes). These positions are placed in a virtual map, such as the one presented in Figure 8. Then, each node is elected as either CM or CH, with probability 1−p and *p*, respectively, at the beginning of the street. This role can change while pedestrians move along the street. Each CH is depicted in this figure with a red circle with the node at the center. Then, the rest of the nodes (CMs) are associated with the closest CH. (In this case, it is a geometric distance, but, in the practical system, the distance can be approximated by the strength of the CH signal.) At this point, we can determine the connection time of each node in its corresponding cluster.

Building on this, we find the connection time histograms considering all the connection times from all of the videos (all pedestrian trajectories recorded) such as the one presented in Figure 9, which is obtained for a radius of 2.5 m, i.e., the red virtual circle is scaled to be 2.5 m. The histograms represent the frequency or the number of samples in each connection time bin. From this, we can characterize the probability density function as described below.

For each histogram, we determine some statistical parameters, such as the mean ($E\left[X\right]$), standard deviation ($\sigma_{X}$), variance ($\sigma_{X}^{2}$), and Coefficient of Variation (CoV), which are defined as follows [41]:(1)$$\begin{array}{c} E\left[X\right]=\frac{1}{n}\sum_{i=1}^{n}x_{i}, \end{array}$$(2)$$\begin{array}{c} \sigma_{X}^{2}=\frac{1}{n}\sum_{i=1}^{n}{\left(x_{i}-E\left[X\right]\right)}^{2}, \end{array}$$(3)$$\begin{array}{c} \sigma_{X}=\sqrt{\sigma_{X}^{2}}, \end{array}$$(4)$$\begin{array}{c} CoV=\frac{\sigma_{X}}{E\left[X\right]}, \end{array}$$
where *n* is the number of connection times measured in the pedestrian trajectories considering *contagion ranges* of 2.5, 3.0, 3.5, and 4.0 m, and 2, 3, 4, and 5 initial CHs. (We consider that some nodes can enter the facilities as either CH or CM and we want to study the effect of having a different number of CHs on the performance of the mobile clustering protocol.) As a result of this analysis, we obtained a CoV higher than one for all considered cases (CoV=1.7). This information is relevant because a Hyper-Exponential distribution can be used to model these connection times. The mathematical model presented below is based on this fact.

In addition, note that, as the communication range increases, the mean connection time is increased accordingly, which is an expected result since pedestrians remain connected longer times.

### Phase-Type Distributions for Pedestrian Connection Times

Given that all the connection times have a CoV=1.7, we focus on finding the parameters of a Hyper-Exponential distribution. This distribution is obtained using two exponential distributions, where the first one with rate $\mu_{1}$ is selected in probability *p*, and the other one with rate $\mu_{2}$ has the complementary probability 1−p, as shown in the following Figure 10, where each phase represents a random exponentially distributed value with rate $\mu_{1}$ and $\mu_{2}$, respectively.

From this, the probability density function is described by [42]
(5)$$f_{X}\left(x\right)=p\theta_{1}e^{-\theta_{1}x}+\left(1-p\right)\theta_{2}e^{-\theta_{2}x}$$

Then, the mean is given by
(6)$$\begin{array}{cc} E\left[X\right]= & \int_{0}^{\infty }xf_{X}\left(x\right)dx \end{array}$$
(7)$$\;\;\;\begin{array}{cc} = & p\frac{1}{\theta_{1}}+\left(1-p\right)\frac{1}{\theta_{2}}, \end{array}$$
and the variance can be calculated, after some algebraic manipulation, as
(8)$$\begin{array}{cc} \sigma_{X}^{2}= & \int_{0}^{\infty }{\left(x-E\left[X\right]\right)}^{2}f_{X}\left(x\right)dx \end{array}$$
(9)$$\;\;\;\;\;\;\begin{array}{cc} = & \frac{p}{\theta_{1}^{2}}\left(2-p\right)+\frac{\left(1-p\right)}{\theta_{2}}\left(\frac{\left(1+p\right)}{\theta_{2}}-\frac{2p}{\theta_{1}}\right). \end{array}$$

Solving (7) for *p*, we get
(10)$$p=\frac{\left(E\left[X\right]-\frac{1}{\theta_{2}}\right)\left(\theta_{1}\theta_{2}\right)}{\theta_{2}-\theta_{1}}.$$

Clearly from this expression, we have the following restrictions: $\mu_{2}>\mu_{1}$ and 0≤p≤1, then:$$\begin{array}{c} \frac{\left(E\left[X\right]-\frac{1}{\mu_{2}}\right)\left(\mu_{1}\mu_{2}\right)}{\mu_{2}-\mu_{1}}<1,\\ \mu_{1}+\left(E\left[X\right]\mu_{2}-1\right)\mu_{1}<\mu_{2},\\ \mu_{1}<\frac{1}{E\left[X\right]}. \end{array}$$

Using these expressions, we can find all the parameters of the Hyper-Exponential distribution, namely, *p*, $\mu_{1}$, and $\mu_{2}$, for each pedestrian environment. The fitting distributions are shown in Table 1, Table 2, Table 3 and Table 4.

## 6. Mathematical Model

We model the potential contagion risk through the time and distance that a node was in contact with another node that later was proven to be positive for SARS-CoV-2 or any other virus that propagates in the air and through the close interchange between people in a pedestrian case. To this end, we derive and numerically solve a Continuous Time Markov Chain (CTMC), depicted in Figure 11, where the states represent the number of nodes inside a mobile cluster. As shown above, the times inside a mobile cluster can be modeled using hyper-exponentially distributed random times. As such, nodes can experience times inside a cluster exponentially distributed either with rate $\mu_{1}$ (and probability *p*), or with rate $\mu_{2}$ (and probability 1−p). The values of these parameters are shown in Table 1, Table 2, Table 3 and Table 4 depending on the particular scenario (number of initial CHs and contagion radio).

Building on this, each state of the proposed CTMC is formed as an order pair $\left(i,j\right)$, where *i* is the number of nodes in phase 1 (nodes enter this phase with probability *p*) and *j* is the number of nodes in phase 2 (nodes enter this phase with probability 1−p). Nodes remain in this phase during the time inside the cluster until they leave it or the cluster vanishes. Hence, the valid states space is described by $\left\{\Omega_{\left(i,j\right)}|0\leq i\leq n_{\max};0\leq j\leq n_{\max}\right\}$. For the specific pedestrian scenarios considered in this work, the number of people in close interactions was not higher than 10. As such, we observe that $n_{\max}=10$, which is the maximum number of nodes inside a mobile cluster observed during the real pedestrian trajectories, but the Markov chain is not limited to this number. In fact, $n_{\max}$ would take the proper value according to the system conditions, i.e., the values of *p*, λ, $\mu_{1}$, and $\mu_{2}$, which is also true for our analysis. In addition, we calculated the rate λ at which nodes can enter an already formed mobile cluster, according to the specific speed and trajectory of each recorded pedestrian.

From this, we can see that transitions in the Markov Chain occur as follows:When the system is in the state $\left(i,j\right)$ ($0<i,j<n_{\max}$), it moves to the state $\left(i+1,j\right)$ when a new arrival to phase 1 occurs with rate p×λ (in case of arrival, i.e., a node entering a particular mobile cluster in phase 1); it moves to the state $\left(i,j+1\right)$ when a new arrival happens to phase 2 with rate $\left(1-p\right)\times \lambda$ (in case of an arrival to phase 2); the system goes to state $\left(i-1,j\right)$ with rate $i\times \mu_{1}$ when a node in phase 1 leaves the cluster; and to state $\left(i,j-1\right)$ with rate $j\times \mu_{2}$ when a node in phase 2 leaves the cluster.When the system is in the state $\left(0,j\right)$ ($0<j<n_{\max}$), the system moves to the state $\left(1,j\right)$ when a new arrival to phase 1 occurs with rate p×λ (in case of arrival, i.e., a node entering a particular mobile cluster in phase 1); it moves to the state $\left(0,j+1\right)$ when a new arrival happens to phase 2 with rate $\left(1-p\right)\times \lambda$ (in case of an arrival to phase 2); note that the system cannot transit to the state $\left(-1,j\right)$; and it moves to the state $\left(0,j-1\right)$ with rate $j\times \mu_{2}$ when a node in phase 2 leaves the cluster.A similar behavior like the previous case occurs in the state $\left(i,0\right)$ ($0<i<n_{max}$) with the appropriate modifications.When the system is in the state $\left(n_{\max},j\right)$ ($0<j<n_{\max}$), it cannot move to the state $\left(n_{\max}+1,j\right)$; it moves to the state $\left(n_{\max},j+1\right)$ when a new arrival happens to phase 2 with rate $\left(1-p\right)\times \lambda$ (in case of an arrival to phase 2); the system goes to the state $\left(n_{\max}-1,j\right)$ with rate $n_{\max}\times \mu_{1}$ when a node in phase 1 leaves the cluster and to the state $\left(n_{\max},j-1\right)$ with rate $j\times \mu_{2}$ when a node in phase 2 leaves the cluster.A similar behavior like the previous case occurs in the state $\left(i,n_{\max}\right)$ ($0<i<n_{\max}$) with the appropriate modifications.In the state $\left(0,0\right)$, only arrivals are allowed, and, in the state $\left(n_{\max},n_{\max}\right)$ only departures can occur with the appropriate rates.

Nodes cannot transit from phase 1 (2) to phase 2 (1) due to the nature of the hyper-exponential process, i.e., nodes either remain in the system with rate $\mu_{1}$ or $\mu_{2}$ but not a combination of these rates, as shown in Figure 10. Since this chain corresponds to an irreducible CTMC, we numerically solve it using the rate equalization method to find the stable state probabilities, $\pi_{\left(i,j\right)}$ that represent the probability that the cluster has *i* nodes in phase 1 and *j* nodes in phase 2.

Note that this Markov Chain can model all the dynamics of the system, and we use it to obtain the main system performance parameters. First, the average number of nodes in a cluster is calculated. Recall that the state of the Markov chain depicts the number of nodes in phase 1, *i*, and the number of nodes in phase 2, *j*, then i+j gives the actual number of nodes at each instant with probability $\pi_{\left(i,j\right)}$, which is found numerically solving the Markov Chain. Hence, the average number of nodes can be calculated as:(11)$$\bar{n}=\sum_{i=0}^{n_{\max}}\sum_{j=0}^{n_{\max}}\left(i+j\right)\pi_{\left(i,j\right)}.$$

Then, we calculate the probability that a packet is successfully transmitted from a CM to their respective CH. Since CM only transmits with probability τ after the reception of the beacon packet, the system behaves as a Slotted ALOHA random access protocol [43], where slots have a duration of $T_{b}$ seconds. In view of this, the packet success probability in a beacon period, when there are *i* nodes in phase 1 and *j* nodes in phase 2, can be calculated as:Psuci,j=i1τ1−τi−11−τj+j1τ1−τj−11−τi.

Hence, the average successful transmission probability in the system can be calculated as:(12)$$P_{\mathrm{suc}}=\sum_{i=0}^{n_{\max}}\sum_{j=0}^{n_{\max}}P_{\mathrm{suc}}\left(i,j\right)\pi_{\left(i,j\right)}.$$

Similarly, the probability of idle beacon period, i.e., the probability that no node transmits in the beacon period when there are *i* nodes in phase 1 and *j* nodes in phase 2, can be calculated as:Pidle(i,j)=i01−τi1−τj+j01−τj1−τi.

Hence, the average idle probability in the system can be calculated as:(13)$$P_{\mathrm{idle}}=\sum_{i=0}^{n_{\max}}\sum_{j=0}^{n_{\max}}P_{\mathrm{idle}}\left(i,j\right)\pi_{\left(i,j\right)}.$$

Finally, the packet collision probability can be calculated as:(14)$$P_{\mathrm{col}}=1-P_{\mathrm{idle}}-P_{\mathrm{suc}}.$$

## 7. Design of the RF Device

In this section, we show the design of an RF device that is carried by every pedestrian of our model. We highlight that this device was designed to be reliable, portable, and simple. As such, this device is not to intended to be a highly sophisticated piece of equipment nor to compete with already available commercial solutions. The RF device consists of four main parts: a microcontroller, a transceiver, an antenna, and a battery. Next, we provide details about each of these parts.

### 7.1. Microcontroller Section

The chip ATMega328p is the microcontroller used in the design of the RF device. It is a low-power CMOS 8-bit microcontroller with an advanced RISC architecture. This microcontroller possesses 32 KBytes of in-system self-programmable flash program memory, 1 KByte EEPROM, and 2 KBytes internal SRAM. In addition, this microcontroller provides six sleep modes: idle, ADC noise reduction, power-save, power-down, standby, and extended standby.

The microcontroller is set up in a stand-alone configuration, running with the internal 8 MHz clock, see Figure 12. The SPI interface is used for burning (*flashing*) the microcontroller’s program, and for driving the transceiver. Several ports are available for interfacing sensors for physical variables such as temperature, heart rate, etc., if necessary. For burning the program, we can use well-known utilities such as AVRDUDE, which is available under Linux, Windows, and Mac OS distributions.

### 7.2. Transceiver Section

A transceiver is needed for establishing wireless communications between CMs and CHs. Several commercial options are available such as XBee modules, see Figure 13a, which provide integrated solutions for configuring a wireless network based on the IEEE 802.15.4 networking protocol. These modules work in the 2.4 GHz ISM band, which is highly populated by the radiation of Wi-Fi and Bluetooth devices, microwave ovens, and other devices. The interfering sources increase the packet error rate [44,45,46]; reduce the throughput [47,48]; and induce higher path loss, and fading [49]. A drawback is the impossibility to write custom firmware for specific applications.

Some cheap RF modules allow certain customization for they lack communication firmware. For instance, the transmitter (TX) module of Figure 13b is a Colpitts oscillator that is turned on/off by an electronic switch, resulting in OOK modulation. The receiver (RX) module is a super-regenerative circuit together with an op-amp comparator for detecting digital symbols. These modules consume up to 20 mW and 10 mW [50], respectively, and work in the ISM band of 315 MHz or 433 MHz. A notable drawback is the large amount of source code needed for equipping the wireless link with the essential functionality for establishing reliable communications, thus occupying most of the microcontroller’s program memory.

On the other hand, there are highly configurable transceiver modules and well developed libraries for existing custom firmware. One example is found in the family of chips nRF24 from Nordic^®^, see Figure 13c, which work in the 2.4 GHz ISM band. Another example comes from the chip CC1101 from Texas Instruments^®^, which work in the sub-1 GHz ISM bands [51]. The coverage range of these modules is quite high when using high-gain well-matched antennas at the maximum output power (up to 0 dBm for the nRF24L01+ at 2.4 GHz, and up to 11 dBm for the CC1101 at 915 MHz).

In the present work, we use the chip MRF49XA from Microchip [52] as transceiver, see Figure 13d. It can work in the 433, 868, and 915 MHz ISM bands. We choose the 915 MHz ISM band since, at this frequency, the RF device can be equipped by a small antenna. Moreover, this band is not populated. The chip employs FSK modulation, with a data rate ranging from 1.2 kbps to 256 kbps. The reception has an increased sensitivity of –110 dBm. The transceiver allows different sleep modes for a reduced overall current consumption. All of the above leads to robust enough wireless links to surpass multipath fading and interference.

Some external components and few extra signals from the ATMega328p are needed for designing a completely RF transceiver, see Figure 14a. Configuration is performed via the SPI interface. The RF interface (RFN and RFP pins of the chip) form an open-collector differential output of 9+i77Ω impedance at 915 MHz. This in turn is the input impedance of the balun designed to feed a 50 Ω antenna, see Figure 14b.

In addition, the MRF49XA provides an analog output for determining the strength of the received signal, when the chip works as a receiver. This is the pin RSSIO, which stands for Received Signal Strength Indicator Output. This signal can be connected to any of the ADC ports of the microcontroller, say, the port PC3, see Figure 14c. The digital value of the RSSIO signal can be used for estimating the closeness of another transmitting node, and determine if that node is inside the contagion range.

The output power of a node working as a transmitter will have losses throughout its trajectory until it reaches the receiving antenna. In our case, the transmitting power is set at the lowest value of –17.5 dBm. Losses include the dispersion by the air interface, coupling losses, polarization losses, among many others [53] (§12.3), so that, at the end, the received power will be in the range of –100 dBm to –60 dBm. Determining the exact value of the received power in a radio link is difficult (not to say impossible) so that at most some estimations can be drawn. Nonetheless, by means of the voltage at the RSSIO pin of the transceiver, we can perform certain calibration processes to estimate distances. That is, we can measure the voltage in this pin in function of the distance to the receiving node under normal conditions in the scenario with pedestrians walking towards the entrance of the building or campus. The voltage corresponding to 2 m is used as a threshold. Thus, if the RSSIO voltage is below this threshold. it implies that the node is outside the contagion radio; otherwise, the nodes are effectively close to each other at a distance less than 2 m. This calibration can be performed to other radii for different infectious diseases. This calibration should agree with Figure 14c, in which the input power dBm on the horizontal axis is translated into distance.

### 7.3. Antenna Section

The antenna of the RF device is made of a single strand of 24 AWG wire (0.5106 mm diameter). Its length *ℓ* was experimentally determined by successively shortening the wire up to observing the resonance at $f_{0}=915$ MHz. This was performed with a vector network analyzer (VNA) MS46121B from Anritsu^®^. Resonance is determined from the scattering parameter $s_{11}$, which corresponds to the reflection coefficient Γ=B/A at the input of the antenna. Here, *B* and *A* are amplitudes of the reflected and incident waves at the input port, respectively. The lower the value of $\left|\Gamma \right|$, the smaller the reflected power and the better the coupling of the antenna. This implies that most of the power supplied to the antenna will be radiated as electromagnetic waves. Similarly, in the reception mode, most of the received power will be transferred to the transceiver.

The measured values of $s_{11}\left(\omega \right)$ as the frequency ω=2πf is swept over a given bandwidth are plotted on a Smith chart, see Figure 15a. The central point $s_{11}=0$ of this diagram represents the best coupling. Around this point, indicated by a gray disc in the chart, the coupling is optimal. On the contrary, the outer circle $\left|s_{11}\right|=1$ corresponds to the worst coupling since all of the energy is reflected. We determined two resonant lengths, namely, $\ell_{1}=10.3$ cm and $\ell_{2}=25.8$ cm, and the measured impedances are $Z_{\mathrm{in}}=37.858+i10.101$ Ω and $Z_{\mathrm{in}}=46.405+i7.702$ Ω, respectively. At 915 MHz, the strokes are closer to the central point associated with the $Z_{0}=50$ Ω, impedance at which the balun was designed. Figure 15b shows the same information but in a Cartesian plane. Resonance corresponds to the minimum reached by the curves, which lie in the gray stripe in the figure.

### 7.4. Battery Section

The power consumption $W=I_{T}V_{CC}$ of the RF device is determined from its consumed current $I_{T}$ at the voltage $V_{CC}$ applied at its terminals. We measure the power consumption in the TX, RX, sleep modes. Let $I_{\mu C}$ and $I_{\mathrm{radio}}$ denote the current consumed by the microcontroller and the transceiver, respectively, thereby $I_{T}=I_{\mu C}+I_{\mathrm{radio}}$. Table 5 and Table 6 show the results of the power consumption in TX mode, and RX and Sleep modes, respectively. The measures were taken with the RF device operating in a continuous form. The first column of Table 5 shows the available transmitting powers of the radio chip, being –17.5 dBm and 0 dBm the lowest and the highest available powers, respectively.

According to the results of the tables, at $V_{CC}=3.3$ V, the average current consumption of the microcontroller in TX mode is $I_{\mu C,\mathrm{TX}}=4.37$ mA; in RX mode, it is $I_{\mu C,\mathrm{RX}}=4.1$ mA; and, in sleep mode, it is $I_{\mu C,\mathrm{Sleep}}=3.94$ mA. These values agree with that specified in the datasheet of the ATMega328p [54], namely, $I_{\mu C,\max}=5$ mA @ 4 MHz, and $V_{CC}=3$ V. With respect to the transceiver, the average current consumption is $I_{\mathrm{radio},\mathrm{TX}}=13.6$ mA and $I_{\mathrm{radio},\mathrm{RX}}=12.8$ mA in TX and RX modes, respectively. No substantial changes are observed in TX or RX mode, and these values are below the typical value indicated in the datasheet of about 17 mA. However, a substantial reduction is observed in the sleep mode, with a current consumption of $I_{\mathrm{radio},\mathrm{Sleep}}=552\;$ A.

The necessary energy for transmitting a chain of digital symbols during a time interval $\Delta t_{\mathrm{TX}}$ is calculated as follows [55]
(15)$$E_{\mathrm{TX}}=W_{\mathrm{TX}}\Delta t_{\mathrm{TX}},$$
where $W_{\mathrm{TX}}$ is the power consumption in TX mode (see Table 5). Similarly, the formulas
(16)$$E_{\mathrm{RX}}=W_{\mathrm{RX}}\Delta t_{\mathrm{RX}},$$
(17)$$E_{\mathrm{Sleep}}=W_{\mathrm{Sleep}}\Delta t_{\mathrm{Sleep}}$$
give the necessary energy for receiving a chain of digital symbols during a time interval $\Delta t_{\mathrm{RX}}$, and for keeping the RF device in the sleep mode during a time interval $\Delta t_{\mathrm{Sleep}}$, respectively, where $W_{\mathrm{RX}}$ and $W_{\mathrm{Sleep}}$ are the corresponding power levels (see Table 6).

Assume a data rate of 100 kbps (which is well supported by the transceiver) in both the TX and RX mode, and assume a sequence of 100 Bytes (that take 8 ms at the considered data rate), then the energy consumption in both modes is: (18)$$E_{\mathrm{TX}}=459.36\;J,\;E_{\mathrm{RX}}=446.16\;J,$$
where we have considered a transmission power of –17.5 dBm. The energy consumption of the RF device in the sleep mode during one second is
(19)$$E_{\mathrm{Sleep}}=14.85\;\mathrm{mJ}.$$

These raw data allow for choosing a suitable battery for feeding the RF device.

In this work, we have considered a LiPo (Lithium-ion Polymer) battery, due to the ease charging, and its small dimensions and weight, which contributes to its portability. For instance, the model DTP502535 [56] is a LiPo battery with a rated capacity of 400 mAh @ 3.7 V, see Figure 16a. The RF device can be fed with this voltage (indeed, the device can work well at the maximum rating voltage of 6 V). Moreover, the maximum rated current is 400 mA, which is more than enough for powering the RF device. The capacity of this battery allows approximately 11,598,746 uninterrupted transmissions and 11,941,904 uninterrupted receptions (of the order of $9.5\times 10^{9}$ bits in both cases), and up to 365,432 s (about 101.5 h, a little more than four days) in the sleep mode continuously. A battery with a larger capacity will increase these estimations proportionally.

LiPo batteries, like the one considered, can be charged by inexpensive charging modules such as the Tp4056 module, see Figure 16b, which provides a USB port for plugging standard phone chargers.

The final assembly of the RF device is shown in Figure 17. This image does not show the battery and the charging module. The small size of the device allows keeping it in a small case that could be worn by the pedestrian. The hardware thus designed is not equipped with a real-time clock or a location mechanism. However, this information can be emitted by a beacon located at strategic locations, such as bus stations, subway entrances, parking lots, or any other location close to the entrance of the campus of the university or other important buildings. In this sense, our RF device is able to handle such information, and can transmit it to the sink node when available. Nonetheless, a more sophisticated design can be included with a real-time clock and a location mechanism, but its power consumption will be greater. For the sake of simplicity and lower power consumption, we opt for the RF device as shown in this work.

## 8. Numerical Results

We now discuss some of the relevant results of this study. First, we validate the Markov chain by comparing the average number of nodes per cluster using the real trajectories and the results obtained through the mathematical model. In Figure 18, we present these results for a different number of initial CHs and contagion radii. We can observe that the results of the model closely match the results of the real trajectories, lightly overestimating the number of nodes having a maximum difference of 0.6 nodes in the worst case. Hence, we believe that the model correctly predicts the behavior of the mobile clustering scheme and can be used to design and analyze contact tracing in other scenarios.

We further verify the accuracy of the mathematical model by calculating the packet success probability using Equation (Equation 12) and comparing it to simulation results. The simulation model consists of tracking each individual node and simulating the mobile clusters based on the protocol described in Section 4. Specifically, we choose the initial number of CHs (2, 3, 4, and 5) by randomly selecting the nodes in the system. Then, the selected CHs begin the beacon transmission. At this point, the system can be found in any of the possible operation scenarios previously described. Then, after all the clusters are found with a single CH and multiple CMs, the packet transmission procedure from CMs to CH begins. Whenever a CM successfully transmits its packet, it stops further transmissions. From this, the simulation counts the time where a single transmission occurs in order to find the success transmission probability (labeled as Real Walk) divided by the total operation time and is compared to the analytical results as shown in Figure 19.

From these results, we can see a good match between simulation and analytical results, especially for two and three initial CHs. In the case of four and five initial CHs, the analytical results slightly separate from the simulation results. The rationale of this is that the mathematical model does not consider the cases where clusters are broken and rebuild in case CH leaves and/or enters a different cluster. Even if we propose the appropriate rules for the mobile clustering protocol to react to these cases, the Markov chain only considers a normal operation situation where clusters are formed by a single CH and multiple CMs. However, the simulation considered all the operation time. As such, in the simulation results, the successful time is lower due to this forming and reforming time of clusters. This effect is further accentuated when the contagion range is increased since the CHs cover a higher area and is more probable to have multiple CHs inside a single cluster, causing higher reconfiguring times.

Now that the mathematical model is validated, and the system variables where the model is more accurate are identified, the following figures use the numerical results derived from the Markov chain and Equations (12) and (13). In Figure 20, we present the success, idle, and collision probabilities in a beacon period. From these results, it is important to note that the successful transmission probability achieves a maximum value when τ is in the range of 0.4 to 0.6. For lower values of τ, the idle (collision) probability is too high (low), while, for higher values, the collision (idle) probability increases (decreases). This is true for almost all values of the initial number of CHs and contagion range.

Another interesting observation is that the performance of the system in terms of these probabilities does not vary much for different radii, which can be explained by observing the dynamics of the pedestrian movement. Indeed, we observe that most of the pedestrians are close to each other and tend to remain together during the trajectories, closer than the minimum contagion range. As such, the number of nodes per cluster is not impacted by increasing the contagion range. We believe that this characteristic may be similar in other scenarios where people are walking to the entrance of another facility or building or subway station, where many people remain in the same range for long periods. However, in other cases, such as people walking in a commercial mall or a park, this characteristic may not be present. We leave this research line open for future works.

Finally, in Figure 21, we present the system performance of the mobile clustering scheme compared to the non-clustered system, i.e., where nodes transmit directly to the sink node. In this case, the sink is located at the entrance of the building. Hence, in the non-clustered system, when nodes enter the communication range of the sink node, they transmit with probability τ.

We can see that the mobile clustering scheme entails better performance than the non-clustered system, in terms of higher successful transmission and lower packet collision probability for almost all the cases investigated, proving the effectiveness of the proposed scheme.

## 9. Conclusions

In this work, we develop a mobile clustering protocol to efficiently provide contact tracing information for contagion contention purposes. We frame this work in the context of smart cities where smartphones of the population or specific communication devices actively share their information with a trusted authority. In a smart city environment, nodes and personal communication devices collaborate with the city administration to achieve common goals for the benefit of the residents and population in general. Since personal information (location and health status) is used to detect possible disease-spreading hot spots, the system has to be supervised and managed by a trusted health authority. The aforementioned system is mathematically modeled, studied, analyzed, and verified through simulation results. The model is accurate in most of the presented scenarios and system parameters. Hence, it can be used for system design in different scenarios than the ones presented in this work. In addition, we designed and constructed an RF capable device that could be used in closed communities, such as university campuses, governmental buildings, hospitals, and schools among others. We focused our research on pedestrians entering the university campus in order to have a controlled environment and obtain data in an expedited manner. However, the same methodology can be used for any other environment, such as commercial, leisure, cultural, and sports events, among others. For these other applications, the first step is to characterize the connection times among people given by the mobility pattern in each scenario, i.e., determine the probability density function of the connection times. In our case, we obtained a Hyper-Exponential distribution where all the parameters were obtained through visual observation. However, in a future work, these times can be obtained directly by the data generated by the Radio Frequency devices or the mobile phones performing the contact tracing application. Given these connection times, we can perform a similar mathematical analysis of the system in order to obtain preliminary/theoretical performance metrics by solving the corresponding Markov Chain. It is important to note that the mobile clustering scheme is independent of these connection times and can operate in any mobility scenario, since CHs and CMs are elected in a distributed manner and based on the number of nodes in the neighborhood. In a future work, we intend to determine the performance of the mobile clustering scheme in such alternative scenarios.

The use of these contact tracing tools will be fundamental in the efforts to control and reduce the impact of the COVID-19 pandemic and future pandemic to come. As such, it is important to have the mathematical and hardware tools to design and implement computational tools in a timely manner in emergency cases. As an additional feature of this work, the use of mobile clusters can be extended to many other applications such as traffic control in vehicular networks or vehicle tracing where also specific devices may be required. In smart city applications in conjunction with autonomous driving scenarios, it may provide a valuable tool for safety and data dissemination.

## Figures and Tables

**Figure 1 entropy-23-00326-f001:**
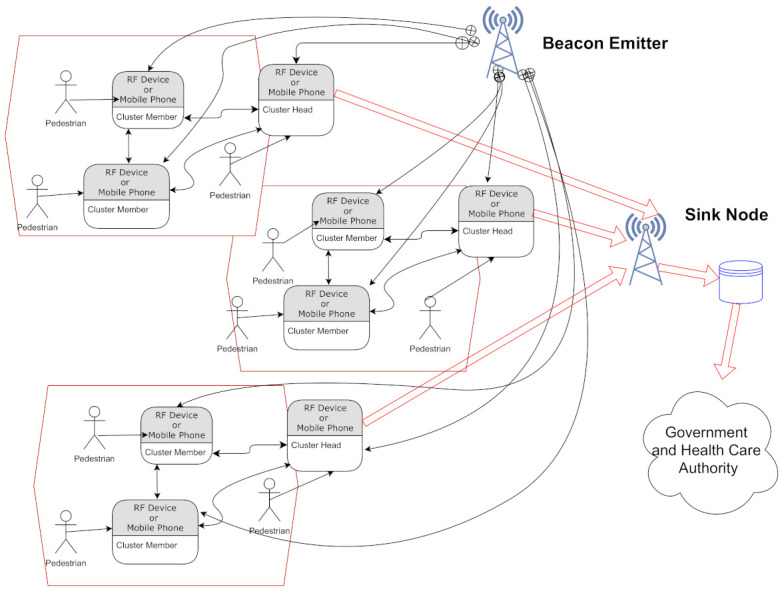
Block diagram of the mobile clustering system.

**Figure 2 entropy-23-00326-f002:**
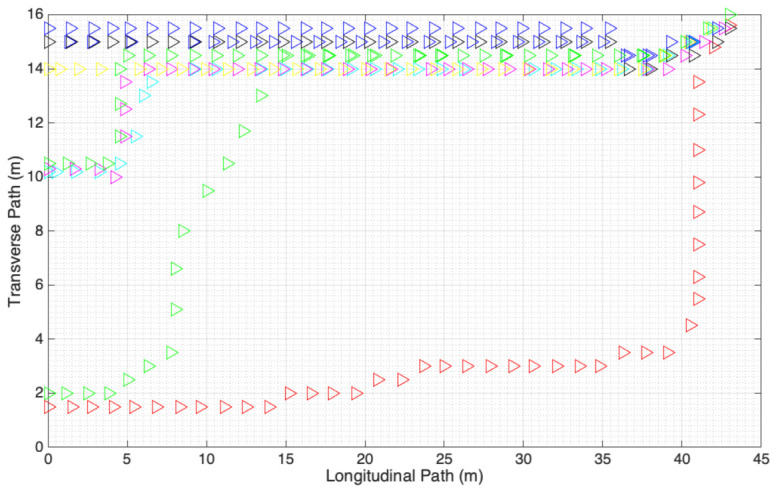
Pedestrian trajectories to access an indoor location.

**Figure 3 entropy-23-00326-f003:**
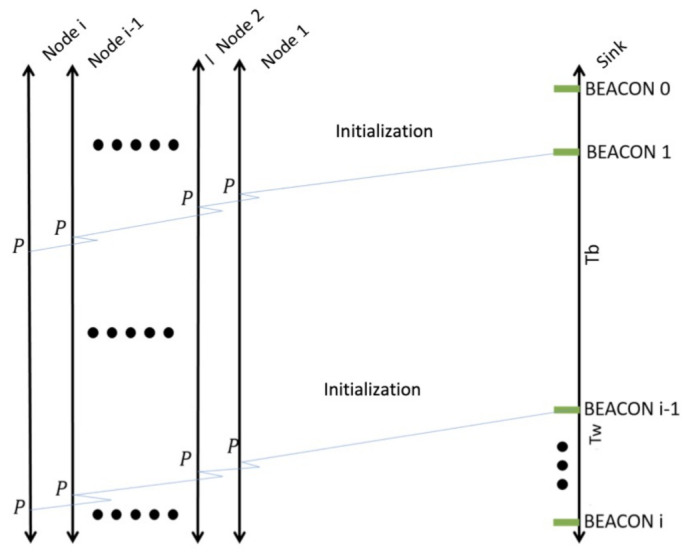
System initialization in a closed community: A beacon signal is transmitted in order for nodes to *wake up* and select their role as either Cluster Head or Cluster Member as well as forming the mobile clusters.

**Figure 4 entropy-23-00326-f004:**
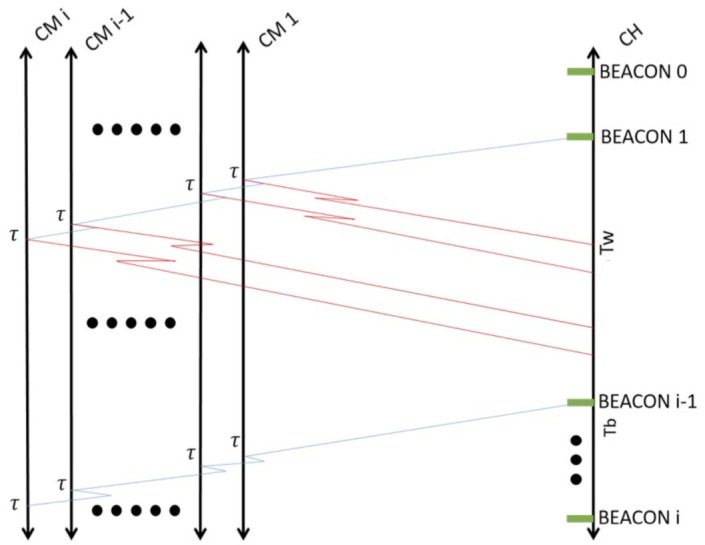
System normal operation: Once nodes have selected their role, Cluster Members transmit their packet to their associated CH which periodically transmit a beacon signal to maintain the clusters.

**Figure 5 entropy-23-00326-f005:**
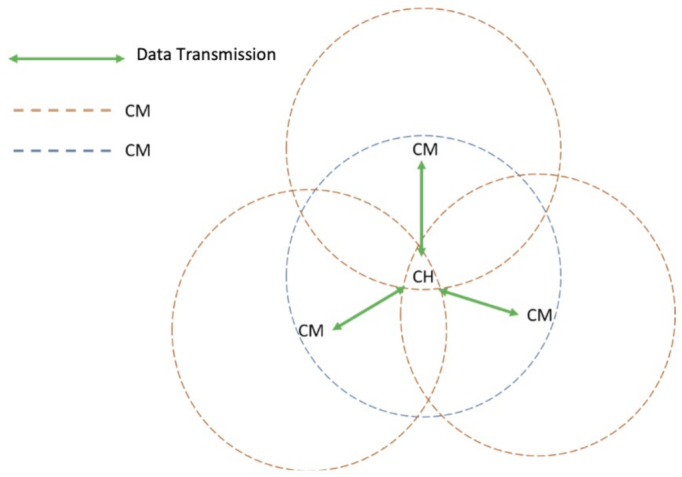
Scenario 1: A single CH and multiple CMs.

**Figure 6 entropy-23-00326-f006:**
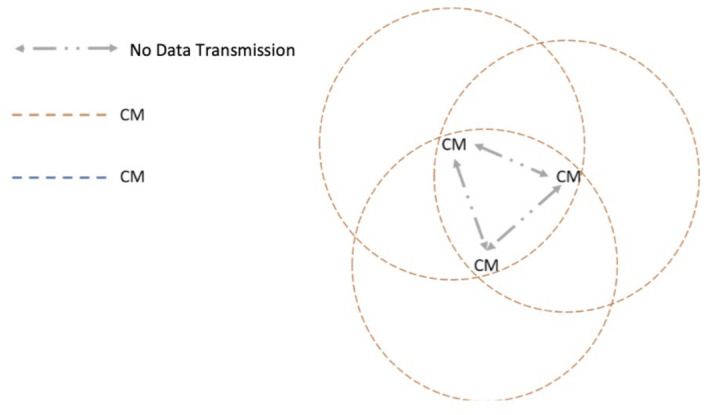
Scenario 2: Multiple CMs and no CH.

**Figure 7 entropy-23-00326-f007:**
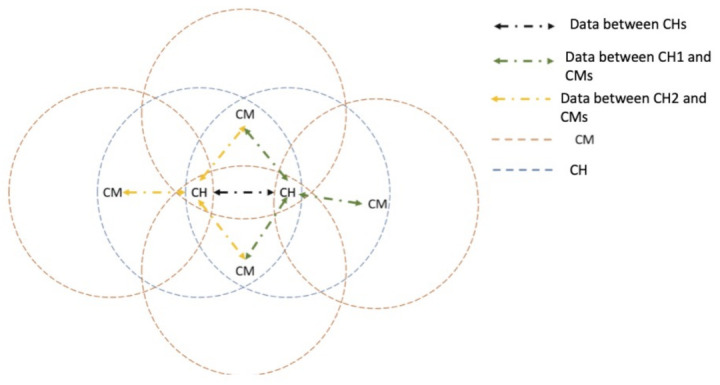
Scenario 3: Multiple CHs and CMs.

**Figure 8 entropy-23-00326-f008:**
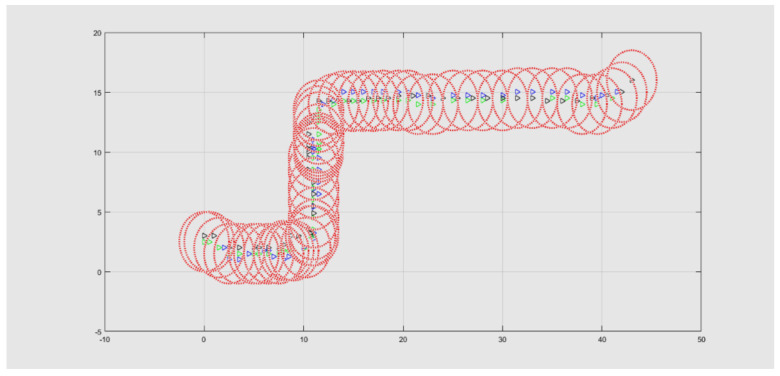
Virtual mobile clusters for the connection radius of 2.5 m.

**Figure 9 entropy-23-00326-f009:**
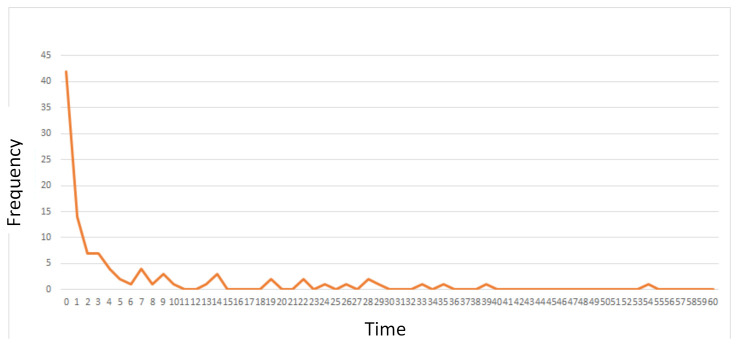
Histogram of the connection times of nodes with a radius of 2.5 m.

**Figure 10 entropy-23-00326-f010:**
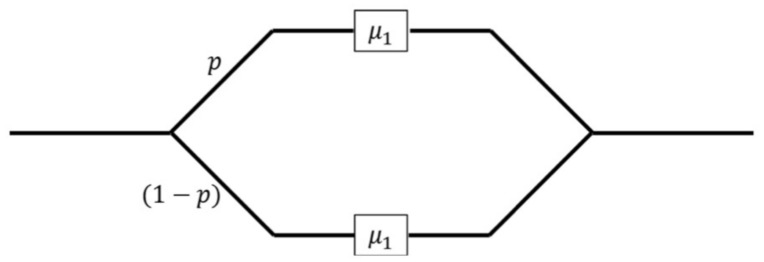
Hyper-exponential process.

**Figure 11 entropy-23-00326-f011:**
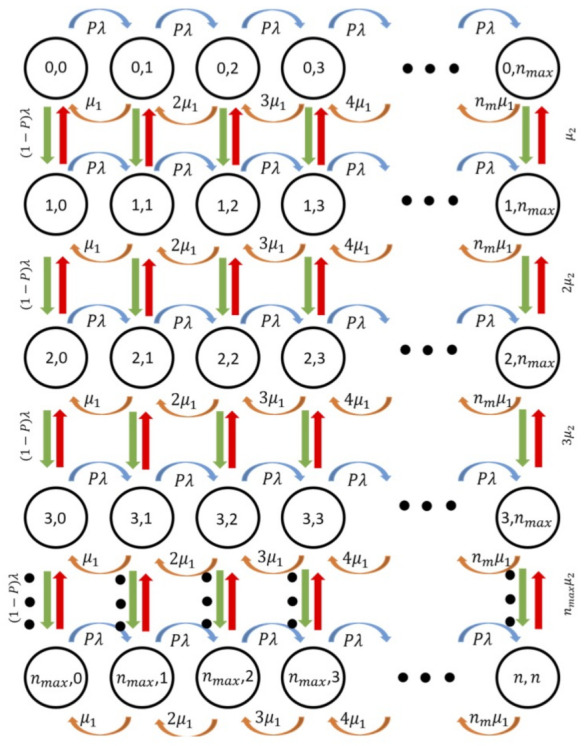
Markov chain that models the mobile clustering scheme.

**Figure 12 entropy-23-00326-f012:**
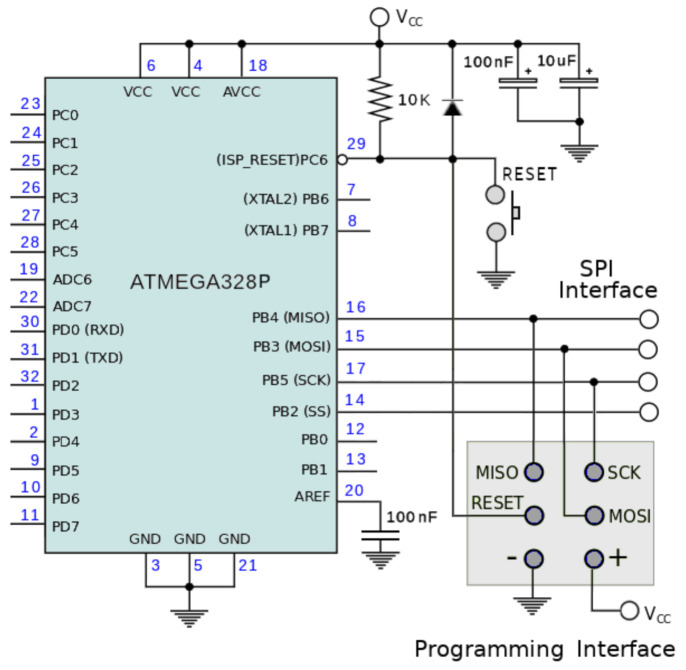
Stand-alone configuration for the ATMega328p.

**Figure 13 entropy-23-00326-f013:**
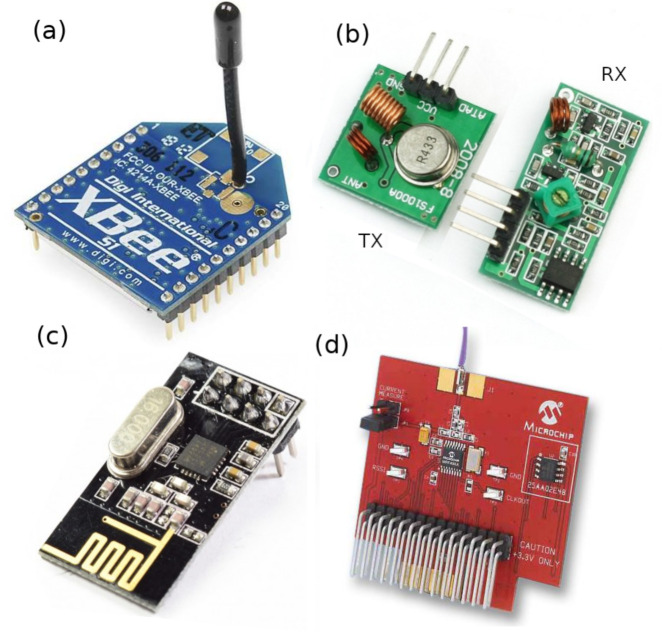
Some commercial RF modules. (**a**) XBee module; (**b**) 433 MHz TX/RX modules; (**c**) NRF24L01+ transceiver; (**d**) MRF49XA transceiver.

**Figure 14 entropy-23-00326-f014:**
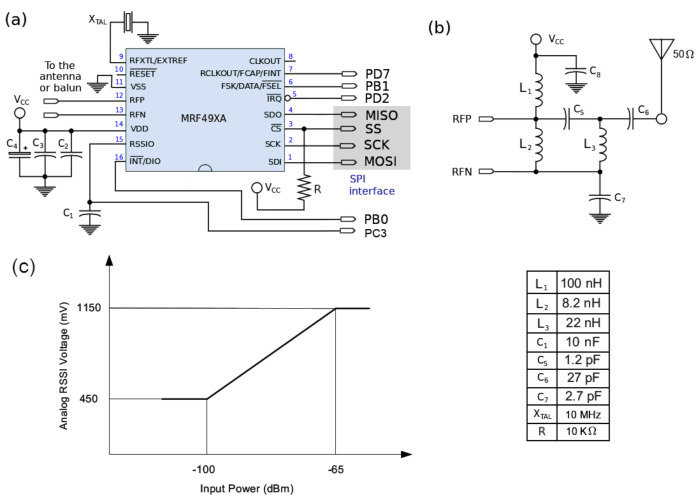
(**a**) Circuitry for the radio chip: $C_{2}$–$C_{4}$, and $C_{8}$ are decoupling capacitors; (**b**) design of the balun for a 50 Ω antenna; (**c**) analog RSSI voltage provided by the chip as a function of power received by the antenna; the larger the receiving power, the shorter the distance of the transmitting node, and vice versa.

**Figure 15 entropy-23-00326-f015:**
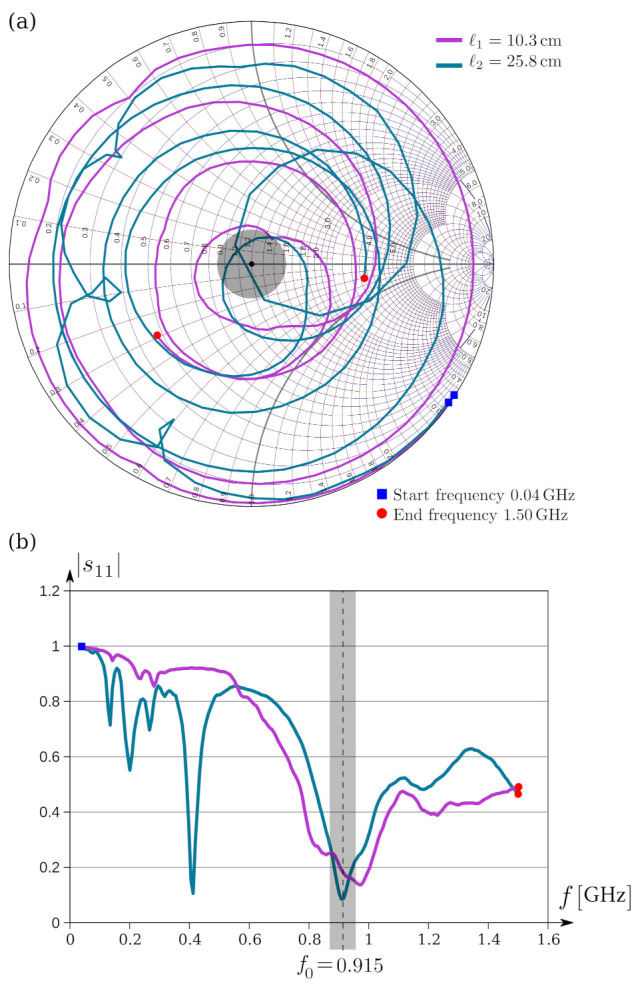
Frequency response of the antenna: (**a**) parameter $s_{11}\left(\omega \right)$ plotted on a Smith chart; (**b**) rectangular plot of $\left|s_{11}\left(\omega \right)\right|$.

**Figure 16 entropy-23-00326-f016:**
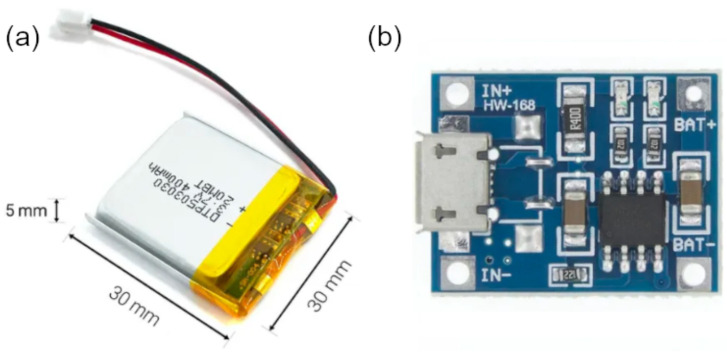
(**a**) LiPo Battery DTP502535; (**b**) charging module Tp4056.

**Figure 17 entropy-23-00326-f017:**
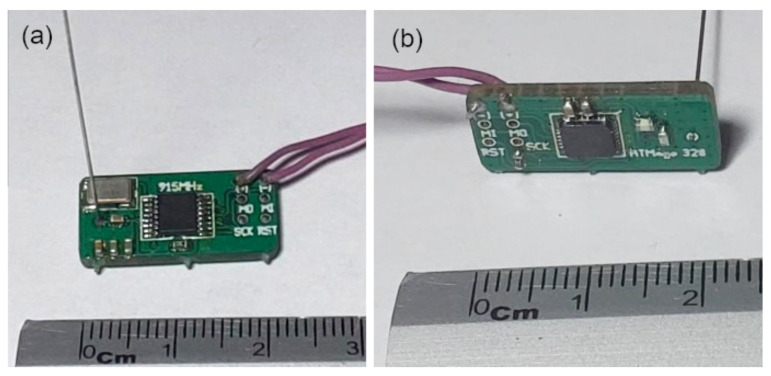
(**a**) Top view of the RF device with the transceiver section and the antenna; (**b**) bottom view of the RF device with the microcontroller section.

**Figure 18 entropy-23-00326-f018:**
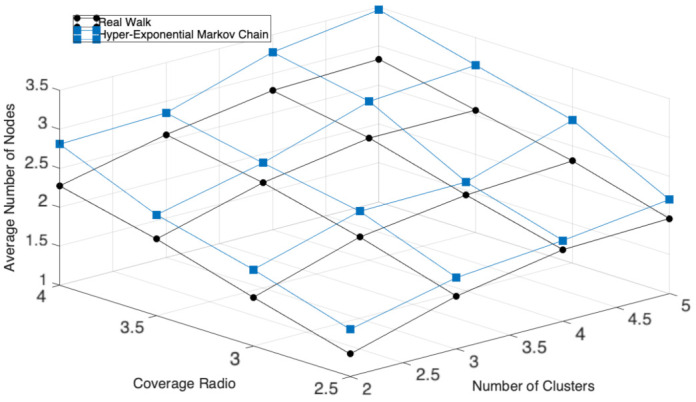
Average number of nodes per cluster: the hyper-exponential approach.

**Figure 19 entropy-23-00326-f019:**
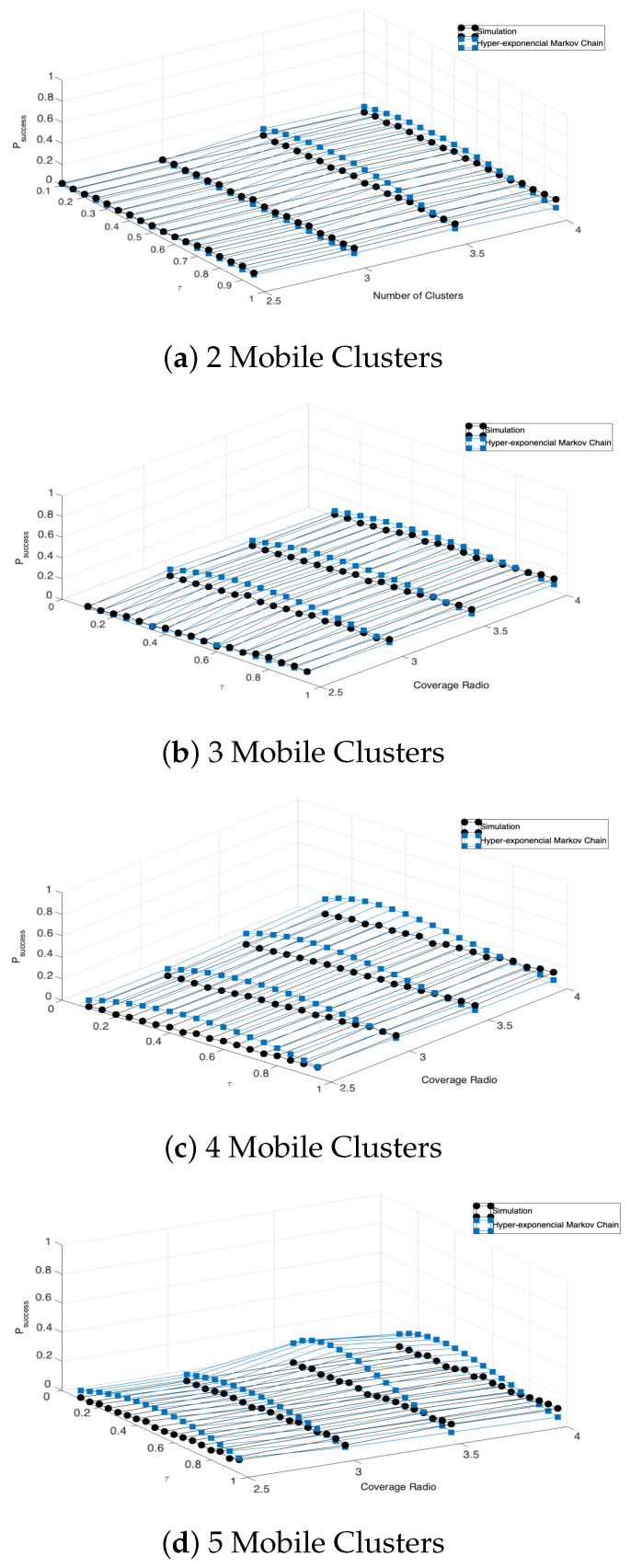
Success probability for the hyper-exponential Markov chain with different number of mobile clusters.

**Figure 20 entropy-23-00326-f020:**
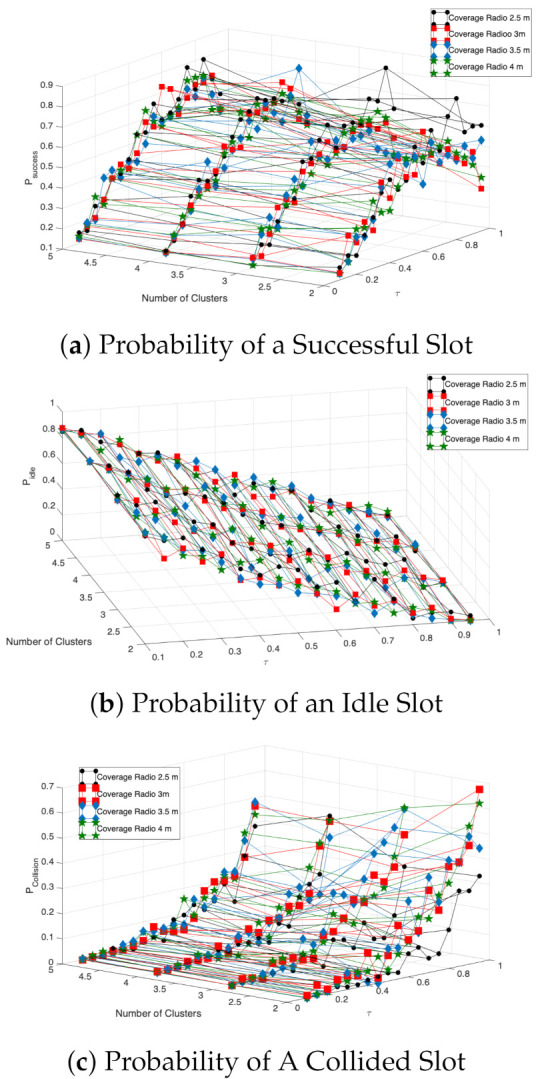
Success, idle and collision probabilities for the mobile clustering scheme.

**Figure 21 entropy-23-00326-f021:**
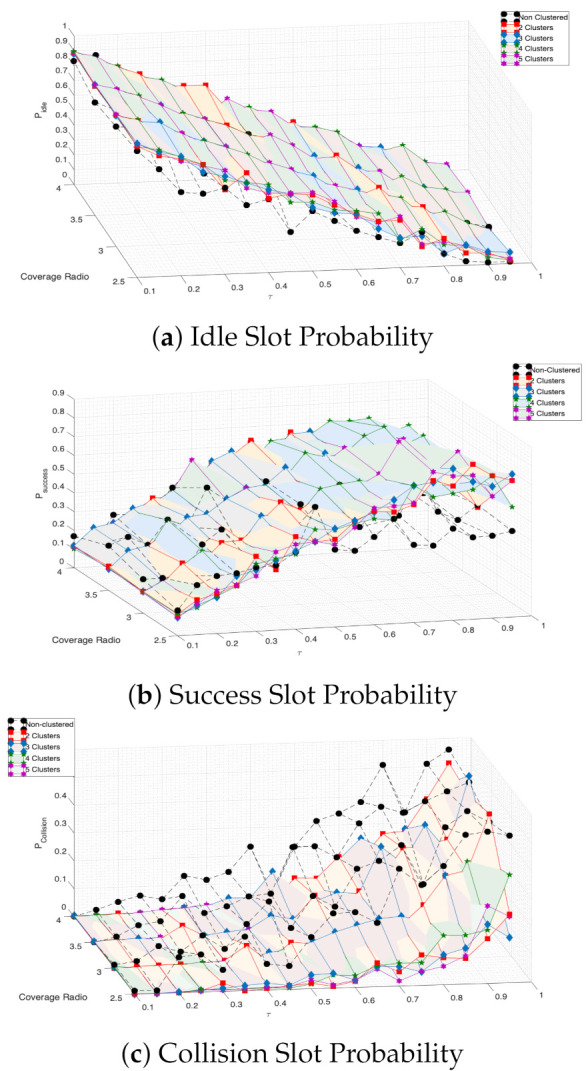
Total registration process: success, idle, and collision probabilities.

**Table 1 entropy-23-00326-t001:** Parameter values of the hyper-exponential distribution for two initial CHs.

Radio	$\mu_{1}$	$\mu_{2}$
2.5	0.078	1.31
3	0.0808	1.33
3.5	0.07	0.2933
4	0.0566	0.168

**Table 2 entropy-23-00326-t002:** Parameter values of the hyper-exponential distribution for three initial CHs.

Radio	$\mu_{1}$	$\mu_{2}$
2.5	0.064	0.2085
3	0.058	0.1321
3.5	0.0608	0.0975
4	0.0722	0.0806

**Table 3 entropy-23-00326-t003:** Parameter values of the hyper-exponential distribution for four initial CHs.

Radio	$\mu_{1}$	$\mu_{2}$
2.5	0.075	1.073
3	0.0589	0.1942
3.5	0.07066	0.1875
4	0.059	0.091

**Table 4 entropy-23-00326-t004:** Parameter values of the hyper-exponential distribution for five initial CHs.

Radio	$\mu_{1}$	$\mu_{2}$
2.5	0.01026	0.1136
3	0.0738	0.1612
3.5	0.079	0.079
4	0.077	0.077

**Table 5 entropy-23-00326-t005:** Power consumption in TX mode at $V_{CC}=3.3$ V.

TX Power	$I_{\mathrm{radio}}$	$I_{\mu C}$	$I_{T}$	*W*
(dBm)	(mA)	(mA)	(mA)	(mW)
0	15.2	4.4	19.6	64.68
−2.5	14.2	4.6	18.8	62.04
−5.0	13.9	4.3	18.2	60.06
−7.5	13.5	4.0	17.5	57.75
−10.5	13.3	4.2	17.5	57.75
−12.5	13.0	4.4	17.4	57.42
−15.0	12.9	4.5	17.4	57.42
−17.5	12.8	4.6	17.4	57.42

**Table 6 entropy-23-00326-t006:** Power consumption in RX and sleep modes at $V_{CC}=3.3$ V.

$I_{\mathrm{radio}}$	$I_{\mu C}$	$I_{T}$	*W*
(mA)	(mA)	(mA)	(mW)
Reception mode			
12.8	4.1	16.9	55.77
Sleep mode			
0.552	3.94	4.5	14.85

## Data Availability

Not applicable.

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
