# Peer review of "Mobile Clustering Scheme for Pedestrian Contact Tracing: The COVID-19 Case Study"

_entropy, 2021, doi:10.3390/e23030326_

Round 1

Reviewer 1 Report

The paper is interesting, structured in a satisfactory way and the
methodological rigor appears adequate to the scientific journal.

The title appears exhaustive in relation to the content of the paper. 

The abstract appears adequate in relation to the entire content of the paper. It contains essential qualitative information. However, no reference is made in it to the appreciable quantitative results obtained. It would be better to include a summary sentence of the numerical results obtained in the abstract. 

The first part of the introduction is well written and concerns information regarding COVID 19 and its transmissibility. The associated bilbiography is interesting and relevant. A good comparison with other forms of influenza was made. The second part of the Introduction was dedicated to the influence of COVID 19 on smart cities. The subject was treated with good methodological rigor and the state of the art was reported in great detail. The bibliography is sufficient and relevant. The description of the proposed approach, in the Introduction, is exhaustive as regards the proposed methodology. However, it should be noted that the data used may be affected by uncertainty and / or inaccuracy. In this regard, it may be necessary to use approaches that exploit soft computing based on fuzzy or neuro-fuzzy techniques. I realize that these approaches would require further developments that could be the subject of future work, so we could limit ourselves, in this paper, to mentioning this possibility by citing specialist works in this sector in the bibliography. To this end, I recommend including the following work in the bibliography:

doi: 10.1155/2014/201243

as it develops an innovative clustering approach using latest generation fuzzy techniques. 

Please make the captions of Figures 2 and 3 self-explanatory. 

It is not clear how (11) was obtained. Please specify better. 

Some formulas used in the paper are not original. Therefore, please associate a relevant bibliographic reference to each of them.

Author Response

The paper is interesting, structured in a satisfactory way and the methodological rigor appears adequate to the scientific journal. The title appears exhaustive in relation to the content of the paper. 

We greatly appreciate the comments and we have made our best effort to correct the manuscript accordingly.

The abstract appears adequate in relation to the entire content of the paper. It contains essential qualitative information. However, no reference is made in it to the appreciable quantitative results obtained. It would be better to include a summary sentence of the numerical results obtained in the abstract. 

We agree, we thank the reviewer for this suggestion. We have added the following paragraph to the abstract:

“Our proposal shows an increased success packet transmission probability and a reduced collision and idle slot probability, effectively improving the performance of the system compared to the case of direct transmissions from each node.”

The first part of the introduction is well written and concerns information regarding COVID 19 and its transmissibility. The associated bilbiography is interesting and relevant. A good comparison with other forms of influenza was made. The second part of the Introduction was dedicated to the influence of COVID 19 on smart cities. The subject was treated with good methodological rigor and the state of the art was reported in great detail. The bibliography is sufficient and relevant. The description of the proposed approach, in the Introduction, is exhaustive as regards the proposed methodology. However, it should be noted that the data used may be affected by uncertainty and / or inaccuracy. In this regard, it may be necessary to use approaches that exploit soft computing based on fuzzy or neuro-fuzzy techniques. I realize that these approaches would require further developments that could be the subject of future work, so we could limit ourselves, in this paper, to mentioning this possibility by citing specialist works in this sector in the bibliography. To this end, I recommend including the following work in the bibliography: doi: 10.1155/2014/201243 as it develops an innovative clustering approach using latest generation fuzzy techniques. 

We agree with the reviewer and we appreciate the future research area suggestion. We have added the following paragraph at the end of Section 4:

“Since our proposal relays on clustering nodes according to the distance among them, and this distance is estimated by the signal strength, there could be inaccuracy and uncertainty given by signal fading, interference and noise in the environment. Hence, to further improve the system precision different fuzzy techniques applied to clustering can be used, such as the one presented in [48].”

Please make the captions of Figures 2 and 3 self-explanatory. 

Done. We have added the following captions:

“System Initialization in a Closed Community: A beacon signal is transmitted in order for nodes to \emph{wake up} and select their role as either Cluster Head or Cluster Member as well as forming the mobile clusters. “

“System Normal Operation: Once nodes have selected their role, Cluster Members transmit their packet to their associated CH which periodically transmit a beacon signal to maintain the clusters.”

It is not clear how (11) was obtained. Please specify better. 

We have added the following paragraph explaining this issue and we have corrected an error in the manuscript in this formula:

“Recall that the state of the Markov chain depicts the number of nodes in phase 1, i, and the number of nodes in phase 2, j, then i+j gives the actual number of nodes at each instant with probability \pi_{(i,j)} which is found numerically solving the Markov Chain. Hence, the average number of nodes can be calculated as”

Some formulas used in the paper are not original. Therefore, please associate a relevant bibliographic reference to each of them.

We added the reference [49] for formulae (1)-(4). For formulae (5)-(9) we added the reference [50]. For formulae (15)-(17) we added the reference [51].

Reviewer 2 Report

Section 2 with related work is too short and lacks details, and needs to be considerably improved. For instance, [2] is referenced as an example of the benefits of contact tracing, but such benefits are not listed nor any details about the study given. Wikipedia provides a much more detailed study of existing contact tracing applications, their benefits and drawbacks in https://en.wikipedia.org/wiki/COVID-19_apps

Although the problem of contact tracing is relevant, the idea proposed in the paper is poorly justified. Designing a new device for people to wear just for tracking contacts due to COVID19 looks like a bad idea. Using applications in a mobile phone is much easier, and has been widely deployed, even though it already raises some significant challenges.
The safety recommendations for a disease like COVID19 are to keep distances between users of at least 2m. This means that clustering makes little sense, particularly if a short range radio is used to save power, like Bluetooth Low Energy (which is probably the best solution for mobile phone contact tracing apps).
The justification to use clusters, in line 76, says "The use of mobile clusters aims at reducing the number of replicated data". Why do data need to be replicated at all?  The smartphone contact tracing application offered by the government in my country does not replicate any data: each mobile phone generates a random ID, and random IDs are exchanged between applications when a risk contact occurs (more than 10 minutes at a distance less than 2 meters); the mobile phone keeps the record of risk contact IDs for 14 days (considered the possible contagion period); if any user is COVID19 positive, it inserts a code provided by the doctor and the app puts its ID in a server for the other users to download and check for possible risk contacts in their smartphone records. So, the app only needs to connect to the database once per day to check new infection cases. There is no need to replicate data or clustering at all.
Having a specific hardware for contact tracing may be a better solution for vehicle tracking, for instance.  

Section 3 describes the system model, but there is no picture showing a block diagram of the system model. From the description, it looks like the system includes a government control agency server (line 118), sink nodes (line 175), cluster heads (CH) and cluster members (CM). Only CH and CM are pictured in later sections.
In line 216, it is said that Cluster Head (CH), receive data (ID, timestamp, location, etc.). How do you get the location and time?  The device designed does not have a real-time clock (RTC) nor any localization mechanism. From lines 164-171, it looks like you are not using Bluetooth signal strength nor GPS, and only identify users within range without any record of the distance. How do you know if another user is within or without the contagion range?
In lines 237-240, it is said that CH transmission range has to be carefully calibrated according to the specific virus characteristics, i.e., the social distance where people can be infected. So this means that if the COVID19 infection distance is set to 2 meters, the transmission range is set to 2 meters?  Have you tried this experimentally? It probably will not work, as the reception error rate at the radio range limit will be too high. What is the purpose of clustering, having clusters with 2 meters?

Section 5 presents mobility statistics, but little details are given on how the data was collected and no interpretation of the results of tables 1-4 is given. Where are the connection time histograms? How can you get connection times from videos?

Why is nmax=10 in line 322?  The number of persons should be unlimited. If the traces have no more than 10 persons at the same time, it looks like the traces are very limited.

Section 7 describes the design of the RF device, describing the different modules, but not how they are assembled together. Are you using an Arduino? Why don't you present a picture of the complete system?

Using the model just for people entering a building looks like a significant limitation of the study. How can the results be generalized for other situations, for instance, people shopping in a supermarket/hypermarket?

Summarizing, the paper needs to be considerably improved before it can be accepted.

Several additional minor corrections are recommended:
- line 107, "needed to kept" -> "needed to keep".
- line 397, "there exist" -> "there are"
- line 398, "for written custom firmware" -> "for existing custom firmware"
- line 406, "an small" -> "a small"
- line 450, the units are missing in the sleep current.
- line 461, replace "LIPO" with "LiPo" and expand the acronym on first use.
- equations are numbered until equation (14), but subsequent equations are not numbered.
- labels in figures 14-17 are too small to be easily read 
- reference [7] has no publication place. You are recommended to show the DOI for all publications that have a DOI.

Author Response

Reviewer 2:

Section 2 with related work is too short and lacks details, and needs to be considerably improved. For instance, [2] is referenced as an example of the benefits of contact tracing, but such benefits are not listed nor any details about the study given. Wikipedia provides a much more detailed study of existing contact tracing applications, their benefits and drawbacks in https://en.wikipedia.org/wiki/COVID-19_apps

We are very grateful with the reviewer for the suggestion. We have added the following new references:

  • https://www.jornada.com.mx/ultimas/capital/2020/11/29/nada-frena-las-aglomeraciones-en-la-cdmx-5571.html
  • https://www.efe.com/efe/america/sociedad/chile-suma-1-699-casos-de-covid-19-con-aglomeraciones-en-la-capital-por-compras-navidenas/20000013-4425665
  • Manawadu, K. L. W. I. Gunathilaka, V. P. I. S. Wijeratne, “Urban Agglomeration and COVID-19 Clusters: Strategies for Pandemic Free City Management”, International Journal of Scientific and Research Publications, Volume 10, Issue 7, July 2020, pp. 769-775. DOI: 10.29322/IJSRP.10.07.2020.p10385
  • https://www.economicsobservatory.com/why-has-coronavirus-affected-cities-more-rural-areas
  • https://en.wikipedia.org/wiki/$COVID-19_apps$
  • https://covid19.who.int/
  • “Contact Tracing in the Real World | Light Blue Touchpaper”. Available: https://www.lightbluetouchpaper.org/2020/04/12/contact-tracing-in-the-real-world/, retrieved 2020-04-15.
  • ”Tracetogether”. Singapore Government. 2020-07-02. Available: https://www.tracetogether.gov.sg/, retrieved 2020-07-02.
  • Bhardwaj N, Khatri M, Bhardwaj S K, Sonne C, Deep A, Kim K. A review on mobile phones as bacterial reservoirs in healthcare environments and potential device decontamination approaches. Environmental Research 2020, 186: 109569. DOI:1016/j.envres.2020.109569

On the other hand, we used the study presented in [2] in order to justify the need of contact tracing and we have added a small description of such study in Section 2. Also, we added the suggested reference in order to explain in more detail the specifics of the contact tracing applications also in section 2. Building on this, the following paragraphs are added in Section II:

“For example, the study in [2] describes the strict contact tracing scheme used in South Korea that uses data from the Global Positioning System (GPS), credit card transactions and video surveillance among other systems in order to reduce the contagion cases of COVID-19, clearly showing the benefits of such policies. “

“Regarding the digital contact tracing tools, there are many different applications with government support in certain regions as mentioned in [9], are centralized protocols concentrating all personal data (geo-localization) in state institutions. For instance, Israel approved the secret service to use surveillance measures to access information of users connected at different networks, which can have many potential privacy issues. Decentralized protocols like the one developed by Covid Watch, the CEN Protocol, based on Bluetooth Low Energy (BLE) using proximity among cellular phones to detect potential contagion cases. In this regard, the Pan-European Privacy-Preserving Proximity Tracing (PEPP-PT) project (a combination of centralized and decentralized approaches) developed a BLE app aimed at detecting such close interactions and avoid state surveillance activities. Later on, different institutions criticized the PEPP-PT for lack of transparency and privacy issues [9]. Nonetheless, these decentralized approaches aim at protecting private information using anonymous keys that have no relation to the user’s identities. However, these applications do not function properly if only a small population uses the app [11], which occurs even if workers are legally required to use it [12]. However in a closed environment such as universities or hospitals where employee access to the buildings can be conditioned on using a specific RF device just like ID is commonly required (or even IDs can be placed on such devices) our proposed device could be a better option since it does not require administrative access to the mobile phone in order to implement contact tracing, and the exposure of smartphones is avoided, which can be potentially dangerous to people since mobile phones are reservoirs for various pathogens [52]. Also, apps that use Bluetooth and GPS to estimate the distance may over-report interactions leading to a high number of false positives [9]. By contrast, the development of a specific device has the advantage of fine-tuning the contagion range according to the specific needs thorough the careful design of antennas, amplifiers, and filters. Indeed, for COVID-19 the official recommendation is to avoid close contact of less than 1.5 or 2 meters, but variations of this virus or for other viruses in the future, this social distancing can be different, and the RF device can be designed accordingly, while GPS and Bluetooth systems cannot easily do. In this work, we propose the use of both approaches, based on apps on mobile phones and specific RF devices, in order to offer a general solution for contact tracing efforts in the sense that the mobile clustering scheme provides an efficient data reporting in pedestrian environments.”

Although the problem of contact tracing is relevant, the idea proposed in the paper is poorly justified. Designing a new device for people to wear just for tracking contacts due to COVID19 looks like a bad idea. Using applications in a mobile phone is much easier, and has been widely deployed, even though it already raises some significant challenges.

We agree with the reviewer that the use of the mobile phone is much easier than a device dedicated to contact tracing. However, as mentioned in [1], there are many privacy concerns related to these applications, especially about systems based on tracking the geographical location of app users. This is one of the main deterrents for installing such tracking systems in personal mobile phones, especially in countries where governmental entities are not trusted by the general public. Such is the case in Latin America and Africa (note that there are no applications listed in [1] for Mexico, Guatemala, Honduras, Angola, Egypt, and many more countries), where also the CPVID-19 epidemic has the highest contagion levels [2]. As such, applications in mobile phones are not a practical solution to keep contact tracing information in all countries. For these cases, the use of a personal device provided by the university or factory or the hospital, or any other place where a high concentration of people is expected, with no other information than a specific ID could be a much-accepted solution. It would also reduce cyber-attacks aimed at obtaining personal information from mobile phones. Hence, we argue that the use of specific devices may not be adopted as a general solution for contact tracing at country level, but it may provide an effective and accepted solution for specific enterprises, commerce locations and governmental and health entities that would allow an anticipated reopening solution to mitigate the economic and social negative impact of generalized lockdowns. For countries where mobile phone applications are well accepted, the mobile clustering scheme and results can be relevant and provide further insights on the performance of such solutions.

We have included a paragraph (highlighted in red) in Section I regarding this issue.

[1] https://en.wikipedia.org/wiki/COVID-19_apps

[2] https://covid19.who.int/

The safety recommendations for a disease like COVID19 are to keep distances between users of at least 2m. This means that clustering makes little sense, particularly if a short range radio is used to save power, like Bluetooth Low Energy (which is probably the best solution for mobile phone contact tracing apps).

It is true that the health-related guidelines are clear aimed at reducing contagion by maintaining a social distance of at least 2 meters. Sadly, these recommendations have not been respected in many documented cases throughout the world [1], [2]. This is the main reason for the increment in confirmed cases at different periods of this pandemic.  In some cases, the 2 meters social distance has not been respected due to cultural, religious, social, and/or commercial reasons where people continue to gather in special dates that they are used to celebrate, despite police installing sanitary filters and trying to separate people in certain regions, as reported in [1].  Furthermore, as mentioned in [3], urban agglomerations, where the social distancing is not fully respected, are the centers at highest risks during a period of a pandemic. Indeed, more than 90% of COVID-19 clusters are associated to densely populated urban agglomerations and megacities in the world [3]. This is can be explained in part by the fact that people have not respected the 2 meters distancing on their daily activities since cities are the economic and financial motor of many developing countries and many activities cannot be completely stopped, workers have to travel, in many cases using the public transport system where close contacts cannot be avoided as noted in [4].  Building on this, we believe that using a shor- range (lower than 2 meters) clustering scheme for communication among pedestrian’s devices for contact tracing applications entail important benefits.

Although ad-hoc wireless sensor networks could be designed based on Bluetooth technology (see, e.g., [53]), we find some issues that limit its application to the tracking of pedestrians in a pandemic scope. In the first place, its high power consumption: establishing a wireless Bluetooth link at a distance of 10m (Power Class 2) or 1m (Power Class 3) requires 4 dBm or 0 dBm, respectively. In this work, we designed a wireless device that employs only -17.5 dBm for establishing a reliable wireless link, which is far lower than the previous transmitting powers. In the second place, the pairing of devices: for the correct pairing the Bluetooth devices should be close enough depending on their transmission power and the level of security. It is observed that the higher the level of security, the larger the average time needed to establish a link (see, e.g., [54]). This may require a few seconds, which is far larger than the microseconds used to transmit data between our RF devices. In the third place, security: though Bluetooth employs an authentication protocol to connect two devices, as well as encryption for transmitting data, it is not difficult to attack a Bluetooth device (see, e.g., [55]), which compromises the protection of information of the pedestrians.

We have added a pair of paragraphs related to this issue in Section I.

[1] https://www.jornada.com.mx/ultimas/capital/2020/11/29/nada-frena-las-aglomeraciones-en-la-cdmx-5571.html

[2] https://www.efe.com/efe/america/sociedad/chile-suma-1-699-casos-de-covid-19-con-aglomeraciones-en-la-capital-por-compras-navidenas/20000013-4425665

[3] L. Manawadu, K. L. W. I. Gunathilaka, V. P. I. S. Wijeratne, “Urban Agglomeration and COVID-19 Clusters: Strategies for Pandemic Free City Management”, International Journal of Scientific and Research Publications, Volume 10, Issue 7, July 2020, pp. 769-775. DOI: 10.29322/IJSRP.10.07.2020.p10385

[4] https://www.economicsobservatory.com/why-has-coronavirus-affected-cities-more-rural-areas

[53] O. Javeri, A. Jeyakumar. Wireless Sensor Network Using Bluetooth. In: S. Unnikrishnan, S. Surve, D. Bhoir (eds) Advances in Computing, Communication and Control. ICAC3 2011. Communications in Computer and Information Science, vol 125. Springer, Berlin, Heidelberg. DOI: 10.1007/978-3-642-18440-6_54

[54] S. Gajbhiye, S. Karmakar, M. Sharma, S. Sharma. Bluetooth secure simple pairing with enhanced security level. Journal of information security and applications, 2019 44: 170–183. DOI: 10.1016/j.jisa.2018.11.009

[55] F. L. Wong, F. Stajano, J. Clulow. Repairing the Bluetooth Pairing Protocol. In: B. Christianson, B. Crispo, J. A. Malcolm, M. Roe (eds) Security Protocols. Security Protocols 2005. Lecture Notes in Computer Science, vol 4631. Springer, Berlin, Heidelberg. DOI:10.1007/978-3-540-77156-2_4

The justification to use clusters, in line 76, says "The use of mobile clusters aims at reducing the number of replicated data". Why do data need to be replicated at all?  The smartphone contact tracing application offered by the government in my country does not replicate any data: each mobile phone generates a random ID, and random IDs are exchanged between applications when a risk contact occurs (more than 10 minutes at a distance less than 2 meters); the mobile phone keeps the record of risk contact IDs for 14 days (considered the possible contagion period); if any user is COVID19 positive, it inserts a code provided by the doctor and the app puts its ID in a server for the other users to download and check for possible risk contacts in their smartphone records. So, the app only needs to connect to the database once per day to check new infection cases. There is no need to replicate data or clustering at all.

We thank the reviewer for pointing this out. We completely agree with this observation. Hence, we have modified the statement of line 76 of the original manuscript (line 114 of the revised manuscript). Also, we leave the “replicated data” remark only for specific devices, acknowledging that for mobile applications this is not an issue, but for specific devices with much lower computational and storage resources that cannot access databases or maintain many contact IDs for many days, it may be an issue.     

Having a specific hardware for contact tracing may be a better solution for vehicle tracking, for instance.  

Yes absolutely. We have added a small paragraph highlighting this research area for future works in the Conclusions section.

Section 3 describes the system model, but there is no picture showing a block diagram of the system model. From the description, it looks like the system includes a government control agency server (line 118), sink nodes (line 175), cluster heads (CH) and cluster members (CM). Only CH and CM are pictured in later sections.

We have added the next block diagram and paragraph depicting the general operation of the system: Please see the attached file.

The system operates by clustering neighbor nodes (associated with pedestrians) moving with a certain trajectory. Given the own dynamics of the users, nodes select their role as either Cluster Head or Cluster Member according to the protocol described in Section 4. The nodes begin the packet transmission after an initialization packet is received which is periodically transmitted by a beacon localized at strategic points. In our case, the beacon emitter is placed at the street leading to the entrance of the National Polytechnique Institute (left part of Figure 2). (For other applications beacon and sink nodes can be placed at the entrance of the supermarket or the subway or hospital, among others.) While pedestrians are walking along the street, they form clusters (depicted by red hexagons) and act according to their role (which can also change according to the possible scenarios described below), i.e., CMs transmitting their packet to their associated (nearest) CH. The approximated distance is calculated by the strength of the signal emitted by each CH. When the CHs detect the sink node, they transmit the gathered information while the CMs remain silent after successful transmission to their respective CH. The sink node stores this information (time and place that the nodes that were part of a cluster and hence, were in close interaction among each other and are potentially in danger of contagion in case that one of them gives positive in a COVID-19 test in the following days) that can be accessed by the trusted authority, e.g., government or health care institution, such that in case of a positive test of the virus, the people in potential contagion danger can be prevented and put in quarantine. In the case of mobile phone users, the sink and beacon emitter function can be performed by the attending base station.

In line 216, it is said that Cluster Head (CH), receive data (ID, timestamp, location, etc.). How do you get the location and time?  The device designed does not have a real-time clock (RTC) nor any localization mechanism.

We have detailed this issue in the new version of the manuscript, and we have added the following paragraph: “For smartphones, this information is available for most cases, while for specific communication devices, this information can be provided by specific beacons placed in strategic locations informing the location and time that the user crossed certain area. In the specific case of the National Polytechnic Institute, these beacons can be placed at the exit of bus stops and subway stations closest to the different campuses or at the streets leading to the entrance of the facilities, like the left part of Figure 1.”

Moreover, at the end of Section 7 we included the following paragraph: “The hardware thus designed is not equipped with a real-time clock, nor a location mechanism. However, this information can be emitted by a beacon located at strategic locations, such as bus stations, subway entrances, parking lots, or any other location close to the entrance of the campus of the university or other important buildings. In this sense, our RF device is able to handle such information, and can transmit it to the sink node when available. Nonetheless, a more sophisticated design can be included with a real-time clock and a location mechanism, but its power consumption will be greater. For the sake of simplicity and lower power consumption we opt for the RF device as shown in this work.”

From lines 164-171, it looks like you are not using Bluetooth signal strength nor GPS, and only identify users within range without any record of the distance. How do you know if another user is within or without the contagion range?

We kindly thank for this opportune observation. Indeed, it is not necessary to use Bluetooth nor GPS, for the transceiver chip provides an analog output to determine the closeness of another node. All the above is specified by a new paragraph at the end of the Transceiver Section:

“In addition, the MRF49XA provides an analog output for determining the strength of the received signal, when the chip works as a receiver. This is the pin RSSIO, which stands for Received Signal Strength Indicator Output. This signal can be connected to any of the ADC ports of the microcontroller, say, the port PC3, see Fig. 14-(c). The digital value of the RSSIO signal can be used for estimating the closeness of another transmitting node, and determine it that node is inside the contagion range.”

 In addition, we have modified Figure 14 to reflect this change. The new figure is the following (Please see the attached file)

Finally, we also modified the caption of the figure by adding the new part: “(c) Analog RSSI voltage provided by the chip as a function of power received by the antenna; the less the receiving power, the larger the distance of the transmitting node, and vice-versa.”

In lines 237-240, it is said that CH transmission range has to be carefully calibrated according to the specific virus characteristics, i.e., the social distance where people can be infected. So this means that if the COVID19 infection distance is set to 2 meters, the transmission range is set to 2 meters?  Have you tried this experimentally? It probably will not work, as the reception error rate at the radio range limit will be too high. What is the purpose of clustering, having clusters with 2 meters?

We kindly appreciate this observation and we understand the confusion. We would like to say that every radiating source indeed is able to reach even infinite distances eventually. However, as the distance r increases the power detected by the receiver will decrease according to O(1/r), or according to some power of 1/r, depending on the media, interfaces, obstacles, etc., that the radiation passes through. That is, no radiating source is able to limit its coverage by no means, and its radiation will propagate far from the source forever. However, in practical terms, every digital receiver is able to correctly identify digital symbols up to certain extent, which is specified by the sensitivity of the receiver. Sensitivity is a measure of the electric field impinging the antenna working in reception mode needed to start up the converter (that is, the LNA-mixer-detector chain). Sensitivity is measured in dBm, which include the work performed by the electric field on a resistive load and depends on the bandwidth and data rate.  The joint use of the RSSIO pin of the transceiver and its sensitivity can determine if a node is inside the contagion radio. These ideas are explained in the following text that we added to the manuscript at the end of the Transceiver Section:

The output power of a node working as a transmitter will have losses throughout its trajectory until it reaches the receiving antenna. In our case, the transmitting power is set at the lowest value of -17.5 dBm. Losses include the dispersion by the air interface, coupling losses, polarization loses, among many others [56, Sec. 12.3], so that at the end the received power will be in the range of -100 dBm to -60 dBm. Determining the exact value of the received power in a radio link is difficult (not to say impossible) so that at most some estimations can be drawn. Nonetheless, by means of the voltage at the RSSIO pin of the transceiver we can perform certain calibration processes to estimate distances. That is, we can measure the voltage in this pin in function of the distance to the receiving node under normal conditions in the scenario with pedestrians walking towards the entrance of the building or campus. The voltage corresponding to 2 m is used as a threshold. Thus, if the RSSIO voltage is below this threshold implies that the node is outside the contagion radio, otherwise, the nodes are effectively close to each other at a distance less than 2 m. This calibration can be performed to other radii for different infectious diseases. This calibration should agree with Figure 14-(c), in which the ‘input power dBm’ on the horizontal axis is translated into distance”.

Finally, in the case of SARS-CoV-2 virus, the 2m distance should be adjusted depending on the content of small particles in the air, so that up to 10m would represent a secure distance. We have added the following phrase in the lines 45-47 of the corrected manuscript: “However, small particles with viral content may travel in indoor environments, covering distances up to 10 meters starting from the emission sources [42].

The following reference was added:

[56] D. Roddy. Satellite Communications. 3 edn. 2001, McGraw-Hill, Inc.: New York.

Section 5 presents mobility statistics, but little details are given on how the data was collected and no interpretation of the results of tables 1-4 is given. Where are the connection time histograms? How can you get connection times from videos?

We provide more detailed information regarding the data collected and how the videos give information about the mobile clusters. We added two new figures, including the histogram for the connection times for a radius of 2.5 meters and the following paragraph:

Specifically, the videos show frame by frame the position of each pedestrian that walked along the street, giving us the location of each potential node (at this point pedestrians are not equipped by RF nodes). These positions are placed in a virtual map, such as the one presented in the following figure. Then, each node is elected as either CM or CH, with probability 1-p and p respectively, at the beginning of the street. This role can change while pedestrians move along the street. Each CH is depicted in this figure with a red circle with the node at the center. Then, the rest of the nodes (CMs) are associated with the closest CH. (In this case, is a geometric distance but in the practical system the distance can be approximated by the strength of the CH signal.) At this point, we can determine the connection time of each node in its corresponding cluster. (Please see the attached file)

Once that the connection times are calculated, we obtained histograms considering all the connection times from all of the videos (all pedestrian trajectories recorded) such as the one presented in the following figure which is obtained for a radius of 2.5 meters, i.e., the red virtual circle is scaled to be of 2.5 meters. The histograms represent the frequency or number of samples in each connection time bin. From this, we can characterize the probability density function as described below. (Please see the attached file).

Why is nmax=10 in line 322?  The number of persons should be unlimited. If the traces have no more than 10 persons at the same time, it looks like the traces are very limited.

In fact, the value of n_{max} it is unlimited, as seen in the valid state space \Gamma, where n_{max} can take any real integer number. However, we identified this practical value due to the fact that in the recorded pedestrian traces this value was not higher than 10. Hence {n_max}=10 is only an observation rather than a limitation.  We clarify this issue in the revised version of the manuscript.

Section 7 describes the design of the RF device, describing the different modules, but not how they are assembled together. Are you using an Arduino? Why don't you present a picture of the complete system?

Indeed, we are not using Arduino at all, though we use the same microcontroller on which Arduino Uno is based, namely, the ATMega328. At the end of Section 7 we added the following paragraph:

“The final assembly of the RF device is shown in Figure 17. This image does not show the battery and the charging module. The small size of the device lets its keeping in a small case that could be wear by the pedestrian.”

And the referred image is the following: (Please see the attached file)

Using the model just for people entering a building looks like a significant limitation of the study. How can the results be generalized for other situations, for instance, people shopping in a supermarket/hypermarket?

True, we focused our research on pedestrians entering the university campus in order to have a controlled environment and obtain data in an expedite manner. However, the same methodology can be used for any other environment, such as commercial, leisure, cultural and sports events, among others. For these other applications, the first step is to characterize the connection times among people given by the mobility pattern in each scenario, i.e., determine the probability density function of the connection times. In our case we obtained a Hyper-Exponential distribution where all the parameters were obtained through visual observation. However, in a future work, these times can be obtained directly by the data generated by the Radio Frequency devices or the mobile phones performing the contact tracing application. Given these connection times, we can perform a similar mathematical analysis of the system in order to obtain preliminary/ theoretical performance metrics by solving the corresponding Markov Chain. It is important to note that the mobile clustering scheme is independent of these connection times and can operate in any mobility scenario, since CHs and CMs are elected in a distributed manner and based on the number of nodes in the neighborhood. In a future work, we intend to determine the performance of the mobile clustering scheme in such alternative scenarios.

We have added a paragraph related to this issue in the Conclusion section of the revised manuscript.

Summarizing, the paper needs to be considerably improved before it can be accepted.

We tried to answer and correct all of the reviewer’s concerns. We hope that we have done so satisfactorily. After the modifications made, we believe that the manuscript has been greatly improved and we sincerely thank the reviewer for the time and effort in the revision of our work.

 Several additional minor corrections are recommended:- line 107, "needed to kept" -> "needed to keep".- line 397, "there exist" -> "there are"- line 398, "for written custom firmware" -> "for existing custom firmware"- line 406, "an small" -> "a small"- line 450, the units are missing in the sleep current.- line 461, replace "LIPO" with "LiPo" and expand the acronym on first use.  Equations are numbered until equation (14), but subsequent equations are not numbered.- labels in figures 14-17 are too small to be easily read, reference [7] has no publication place. You are recommended to show the DOI for all publications that have a DOI.

We have corrected all of these errors and added the appropriate DOIs to all available references.

Round 2

Reviewer 2 Report

The authors properly addressed the comments to the previous version. 
The paper has been significantly improved over the previous version and can now be considered for publication.

Only a few minor corrections are proposed:
- line 115, CHs is used but only introduced later in line 123
- line 151, replace "CPVID-19" with "COVID-19"
- line 582, replace "determine it" with "determine if"
- line 657, replace "lets its keeping" with "allows keeping it"